# Progerin reduces LAP2α-telomere association in Hutchinson-Gilford progeria

**Alexandre Chojnowski[1], Peh Fern Ong[2], Esther SM Wong[1], John SY Lim[3], Rafidah A Mutalif[1], Raju Navasankari[1], Bamaprasad Dutta[4], Henry Yang[5], Yi Y Liow[1], Siu K Sze[4], Thomas Boudier[6,7], Graham D Wright[3], Alan Colman[8], Brian Burke[9], Colin L Stewart[1,10]\*, Oliver Dreesen[2]\***

[1]Developmental and Regenerative Biology, Institute of Medical Biology, Singapore, Singapore; [2]Cellular Ageing, Institute of Medical Biology, Singapore, Singapore; [3]Microscopy Unit, Institute of Medical Biology, Singapore, Singapore; [4]School of Biological Sciences, Nanyang Technological University, Singapore, Singapore; [5]Bioinformatics Core, Cancer Science Institute of Singapore, National University of Singapore, Singapore, Singapore; [6]Bioinformatics Institute, IPAL UMI 2955, Singapore, Singapore; [7]Image and Pervasive Access Lab, Sorbonne Universites, Paris, France; [8]Stem Cell Disease Models, Institute of Medical Biology, Singapore, Singapore; [9]Nuclear Dynamics and Architecture, Institute of Medical Biology, Singapore, Singapore; [10]Department of Biological Sciences, National University of Singapore, Singapore, Singapore

**\*For correspondence:** colin.
stewart@imb.a-star.edu.sg (CLS);
oliver.dreesen@imb.a-star.edu.
sg (OD)

**Competing interests:** The
authors declare that no
competing interests exist.

**Reviewing editor:** Karsten Weis,
ETH Zürich, Switzerland

**Abstract** Hutchinson-Gilford progeria (HGPS) is a premature ageing syndrome caused by a mutation in *LMNA*, resulting in a truncated form of lamin A called progerin. Progerin triggers loss of the heterochromatic marker H3K27me3, and premature senescence, which is prevented by telomerase. However, the mechanism how progerin causes disease remains unclear. Here, we describe an inducible cellular system to model HGPS and find that LAP2α (lamina-associated polypeptide-α) interacts with lamin A, while its interaction with progerin is significantly reduced. Super-resolution microscopy revealed that over 50% of telomeres localize to the lamina and that LAP2α association with telomeres is impaired in HGPS. This impaired interaction is central to HGPS since increasing LAP2α levels rescues progerin-induced proliferation defects and loss of H3K27me3, whereas lowering LAP2 levels exacerbates progerin-induced defects. These findings provide novel insights into the pathophysiology underlying HGPS, and how the nuclear lamina regulates proliferation and chromatin organization.

## Introduction

The nuclear lamina, a proteinaceous meshwork consisting of A-type and B-type lamins, underlies the inner nuclear membrane and is important for maintaining interphase nuclear architecture. In addition, it provides a structural scaffold for factors involved in DNA repair, replication and transcription (*Burke and Stewart, 2006*; *Dechat et al., 2008*). Mutations in the *LMNA* gene are responsible for a variety of human genetic disorders, collectively called the laminopathies (*Burke and Stewart, 2006*; *Worman et al., 2010*). Laminopathies include forms of muscular dystrophy, cardiomyopathy, lipodystrophy and the premature aging disease, Hutchinson-Gilford Progeria Syndrome (HGPS).

HGPS patients appear normal at birth, but by 12–18 months begin to exhibit features associated with accelerated ageing, and usually die in their teens due to cardiovascular failure. HGPS is caused by an autosomal dominant C to T nucleotide substitution at position 1824 (G608G) in *LMNA*, which activates a cryptic splice site and results in a truncated and constitutively farnesylated version of lamin A called progerin (*De Sandre-Giovannoli et al., 2003*; *Eriksson et al., 2003*). HGPS fibroblasts have a greatly reduced proliferative capacity, abnormal nuclear architecture, persistent activation of DNA

**eLife digest** Hutchinson-Gilford Progeria Syndrome (HGPS) is a rare genetic disease in which individuals age prematurely. Newborns appear normal at birth, but start ageing rapidly when they are around a year old. Symptoms of the disease include stunted growth and joint stiffness, and individuals often die of heart failure during their teens.

A mutated version of a protein called lamin A causes HGPS; this mutant is known as progerin. In cells that produce progerin, the 'telomeres' that protect the ends of chromosomes (the structures that contain most of the cell's DNA) from damage, are unusually short. Every time a cell divides, the telomeres get shorter. If they get too short, the DNA is damaged and the cell stops dividing and enters a state known as senescence.

HGPS affects some of the tissues in the body more severely than others, and these tissues tend to produce high levels of progerin. By gradually raising the levels of progerin in human cells, Chojnowski et al. found that DNA damage and cell senescence only occur when the amount of progerin in a cell exceeds a particular threshold. Moreover, the expression of telomerase—a complex that can elongate telomeres—prevented progerin-induced DNA damage and premature senescence.

To find out how progerin affects cells, Chojnowski et al. compared how lamin A and progerin interact with other proteins. This revealed that progerin interacts with a protein called LAP2α more weakly than lamin A. LAP2α normally associates with telomeres, but using super-high resolution microscopy, Chojnowski et al. observed that this association is less likely to occur in the cells of people with HGPS. Importantly, increasing the amount of LAP2α in progerin-expressing cells prevented DNA damage and senescence and enabled these cells to continue dividing.

Chojnowski et al. propose that in HGPS, the weak interaction between LAP2α and progerin disrupts how LAP2α interacts with telomeres, which prevents cells from dividing. Understanding this process may help to design new ways of treating HGPS, and may also help us to understand other diseases that are caused by mutations in lamin proteins.

damage checkpoints and shortened telomeres (*Allsopp et al., 1992*; *Bridger and Kill, 2004*; *Goldman et al., 2004*; *Liu et al., 2005*, *2006*; *Decker et al., 2009*). Critically shortened telomeres elicit a DNA damage response and trigger senescence, resulting in irreversible growth arrest (*d'Adda di Fagagna et al., 2003*). Previous results revealed that ectopic expression of telomerase reverse transcriptase (TERT) extends the proliferative capacity of HGPS fibroblasts and rescues progerin-induced DNA damage (*Kudlow et al., 2008*; *Benson et al., 2010*). However, it remains unknown to what extent ectopic expression of TERT rescues all progerin-induced phenotypes, and whether physiological levels of TERT are sufficient. It also remains unclear how progerin triggers senescence and why specific tissues in HGPS patients are more affected than others.

Here, we describe a regulatable cellular model of progeria and show that upon induction in primary human fibroblasts, progerin leads to increased DNA damage, cellular senescence, senescence-associated reduction of lamin B1, nuclear morphology defects and altered expression of H3K27me3, in a dose-dependent manner. Exogenous TERT prevents the proliferative defects, DNA damage, lamin B1 reduction and gene expression differences induced by progerin, although nuclear morphology defects and altered deposition of H3K27me3 are not prevented by TERT.

To determine how progerin may induce these defects, we compared the protein interactome between lamin A and progerin. This revealed that the physical association between progerin and the α-isoform of the lamina-associated polypeptide 2 (LAP2α) is disrupted. Increased levels of LAP2α, but not LAP2β, suppressed many of the progerin-induced defects, including the inhibition of cell proliferation and reduction in heterochromatin, revealing that LAP2α plays a central role in the molecular pathology of progeria.

## Results

### Progerin impairs proliferation, induces premature senescence and loss of lamin B1 in primary fibroblasts

To investigate the mechanism of progerin's effects, we developed a tractable experimental system utilizing primary (telomerase-negative) and telomerase-positive (expressing pBABE-Neo-hTERT)

(TERT+) human fibroblasts. These were then either induced to express V5-tagged lamin A or progerin to model the pathophysiology of HGPS fibroblasts. We used a doxycycline (DOX) inducible lentiviral based system to quantitatively induce the expression of progerin and lamin A to levels comparable to those present in HGPS fibroblasts (*Figure 1A,B*; *Figure 1—figure supplement 1A,B,H*). The advantage of such a system is that it accurately tracks the replicative history of isogenic cell lines, removing the uncertainty in using HGPS-patient derived cells where passage number and telomere length may be unknown, as described (*Decker et al., 2009*).

Progerin expression leads to misshapen nuclei (*Figure 1—figure supplement 1B*), inhibition of cell proliferation (*Figure 1C*, *Figure 1—figure supplement 1C*), premature senescence (as measured by expression of senescence-associated β-galactosidase, [SA-β-gal]), a reduction in lamin B1 levels, (*Figure 1—figure supplement 1A,D,G*) and the induction of 53BP-1 DNA-damage foci (*Figure 1D*). In addition, western blot analysis from three independent experiments demonstrated that expression of TERT did not reduce progerin levels (*Figure 1B,E* and *Figure 1—figure supplement 1E*). All of these changes are consistent with previous findings from HGPS and senescent fibroblasts (*Scaffidi and Misteli, 2005*; *Taimen et al., 2009*; *Shimi et al., 2011*; *Freund et al., 2012*; *Dreesen et al., 2013*). The expression of exogenous TERT prevented progerin-induced proliferative defects, loss of lamin B1, and reduced the number of SA-β-gal positive cells to background levels (*Figure 1B,D*; *Figure 1—figure supplement 1A,C,D,G*). Consistent with these results, expression of TERT in HGPS patient derived fibroblasts increased telomere length, restored their proliferative capacity and reduced the number of cells with DNA damage foci (*Figure 1—figure supplement 1H–J*). In contrast, expression of TERT did not ameliorate the nuclear abnormalities of HGPS cells (*Figure 1—figure supplement 2A*). Lastly, expression of normal lamin A did not significantly affect the proliferation rates of primary- or TERT+ human fibroblasts (*Figure 1—figure supplement 1F*).

One of the most intriguing aspects of HGPS and other premature aging syndromes is the tissue-specific manifestation of the disease. Different lineages derived from reprogrammed HGPS induced pluripotent cells (iPSC) expressed varying amounts of progerin, with neural lineages showing the lowest levels, consistent with the fact that they remain unaffected in HGPS patients (*Zhang et al., 2011*). These results suggested that progerin's detrimental effects depend on its levels of expression in a given tissue. To test this, we expressed increasing levels of progerin in primary and TERT+ fibroblasts by increasing the concentration of DOX (0–2000 ng/ml). This resulted in progerin inhibiting the proliferation of primary fibroblasts in a dose-dependent manner (*Figure 1C*). Defective proliferation was accompanied by a gradual increase in DNA damage levels, as quantified by the number of 53BP1 foci per nucleus (*Figure 1D*, left panel, p < 0.01). The levels of progerin required to induce a phenotype corresponded to ∼30–40% of endogenous lamin A levels (at 100–250 ng/ml DOX; *Figure 1B–D*). Progerin therefore must reach a certain threshold to induce DNA damage and inhibit proliferation. Both of these effects were suppressed by exogenous TERT (*Figure 1B–D*, right panel).

Previous studies reported that progerin leads to a decrease in repressive histone marks including H3K27 trimethylation and loss of peripheral heterochromatin (*Shumaker et al., 2006*; *McCord et al., 2013*). To determine whether TERT would prevent progerin-induced chromatin alterations, we measured the levels of progerin and H3K27me3 in single cells by immunofluorescence. As shown in *Figure 1E*, a scatterplot analysis of >9800 nuclei revealed an inverse correlation between progerin and H3K27me3 levels (Pearson r = −0.43 and −0.24 for TERT negative and TERT+ cells expressing progerin, respectively, p < 0.001, *Figure 1E*, inset). However, progerin mediated loss of H3K27me3 was not prevented by TERT. The inverse correlation between progerin expression and loss of H3K27me3 is also apparent in two primary HGPS cell lines (*Figure 1F*, Pearson r = −0.70 and −0.52 for HGPS 01972 and HGPS 11513 respectively, p < 0.001), as well as in one TERT+HGPS cell line (*Figure 1—figure supplement 1K*, Pearson r = −0.5). To determine whether this altered chromatin state was sufficient to affect genome-wide gene expression, and whether TERT would prevent these changes, we performed a microarray analysis on primary and TERT+ cells expressing either lamin A or progerin at 4 weeks after induction. The expression of 142 genes was increased or decreased more than twofold in progerin vs lamin A expressing cells (*Figure 1G*). Many of these gene expression changes were associated with senescence, including a reduction of Wnt2 (*Ye et al., 2007*), increased expression of matrix metalloproteinases (*Kang et al., 2003*) and plasminogen activator inhibitor-1 (PAI-1) (*Kortlever et al., 2006*). Expression of TERT prevented nearly all these changes in the differentially expressed genes, and restored the gene expression profile to that seen in cells not expressing progerin (*Figure 1H*). These results show that the inducible system reliably phenocopies

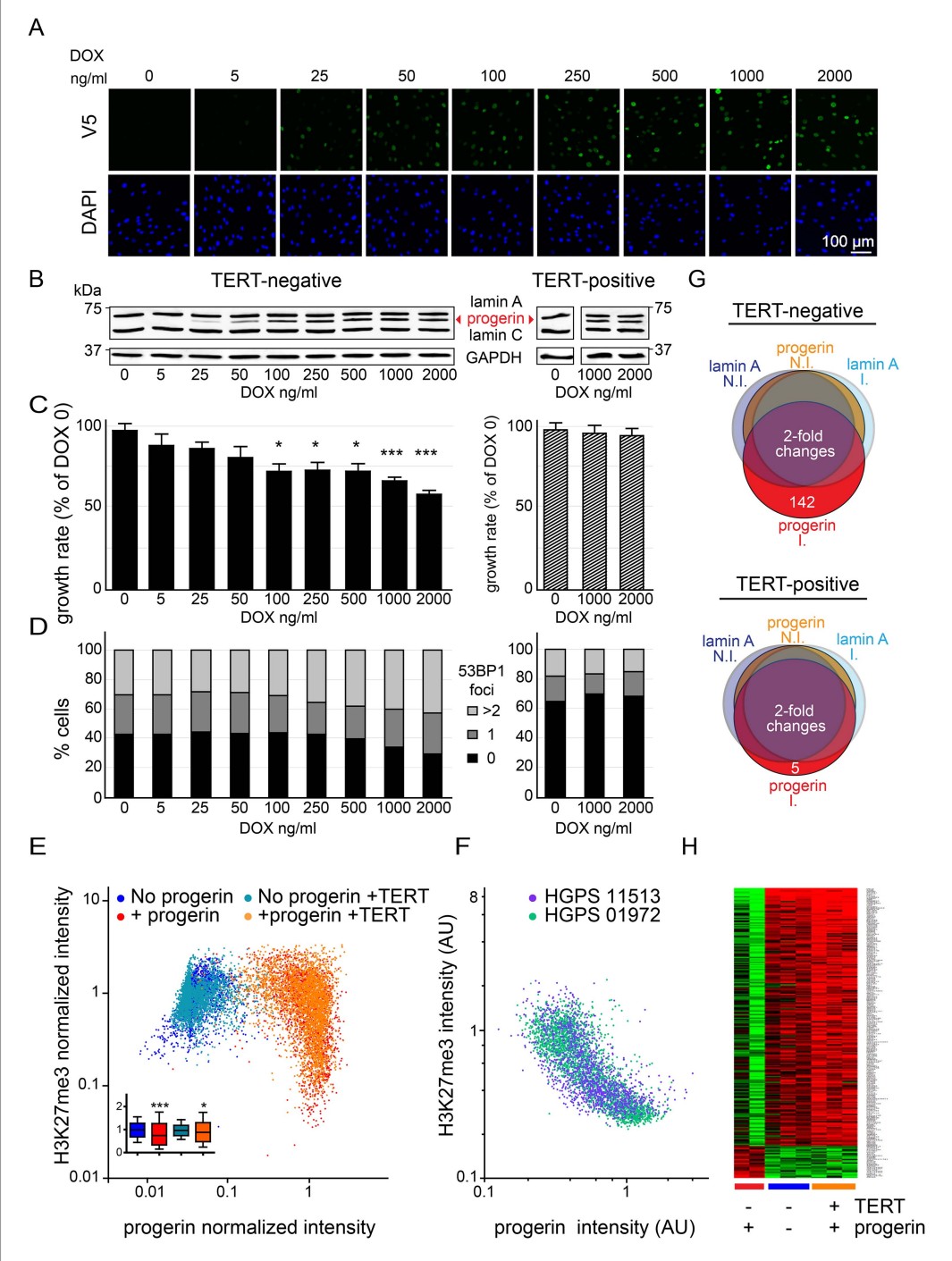

**Figure 1**. Telomerase rescues dose dependent progerin-induced proliferation defects, DNA damage and gene expression changes without alleviating chromatin changes. (**A**) Immunofluorescence microscopy using v5-tag antibody showing doxycycline-dependent inducible expression of v5-progerin and its localization to the nuclear periphery. DAPI staining is shown on the bottom panels. Scale bar: 100 μm. (**B**) Western blotting showing doxycycline-dependent progerin expression in primary (left panel) and TERT+ (right panel) fibroblasts. Progerin migrates between lamin A and C as indicated (red arrowhead). Doxycycline concentrations (0–2000 ng/ml) are indicated under each lane. (**C**) Quantification of progerin-induced proliferation defects. Relative growth rates of primary (left panel) and TERT+ cells (right panel) according to progerin expression levels (*p < 0.05; ***p < 0.001 compared to control 0 ng/ml doxycycline, error bars represent SEM, 2-way ANOVA with Bonferroni's post-test). (**D**) Quantification of progerin-induced 53BP1 DNA damage foci in response to progerin expression levels, in

*Figure 1. continued on next page*

*Figure 1. Continued*

primary (left panel, p < 0.01, $\chi^2$ test) and TERT+ cells (right panel). 350–500 cells were counted for each condition. (**E**) Scatter plot analysis of primary (blue, red) and TERT+ (cyan, orange) cells showing an inverse correlation between H3K27me3 and progerin expression in each cell nucleus using immunofluorescence microscopy (Pearson r = −0.43 and −0.24 for TERT negative and TERT+ cells expressing progerin, respectively, p < 0.001, n > 9800). Inset: box plot of the same data, whiskers represent 10–90 percentile (***p < 0.001, *p < 0.05, one way ANOVA with Bonferroni's post-test). (**F**) Scatter plot analysis of H3K27me3 and progerin expression in single nuclei of two primary HGPS lines using immunofluorescence microscopy (Pearson r = −0.70 and −0.52 for HGPS AG01972 and HGPS AG11513 respectively, p < 0.001, n > 4000). (**G**) Illustration showing the number of genes whose expression changed more than twofold after 28 days of lamin A or progerin expression (I, induced. N.I., non-induced). No significant changes were observed upon expression of lamin A. In primary and TERT+ cells, 5 and 142 genes were differentially regulated upon progerin expression, respectively. (**H**) Heatmap representation of the 142 differentially regulated genes in the presence or absence of progerin and TERT, in human fibroblasts.

The following figure supplements are available for figure 1:

**Figure supplement 1**. Progerin induced senescence, lamin B1 loss, DNA damage, and telomere shortening are prevented by TERT in primary and HGPS fibroblasts, control experiments.

**Figure supplement 2**. Expression of hTERT or LAP2α does not alleviate nuclear abnormalities in HGPS cells.

---

HGPS cell characteristics in isogenic cell lines, and that ectopic expression of TERT prevents dose-dependent DNA damage, premature cellular senescence and senescence-associated changes in gene expression induced by progerin, independent of its impact on H3K27 methylation.

To determine whether endogenous physiological levels of TERT would recapitulate the effects of exogenous TERT expression, we expressed lamin A and progerin in mouse embryonic stem cells (ESC). Endogenous TERT expression is a hallmark of ESCs and enables them to perpetually self-renew. Both lamin A and progerin were expressed in the ESC nuclei upon addition of DOX and localized to the nuclear periphery (*Figure 2—figure supplement 1A,B*). Expression of the exogenous progerin or lamin A did not impair the proliferation of the pluripotent ESCs (*Figure 2A*), induce significant changes in gene expression (*Figure 2B*), alter nuclear lamina structure, as measured by lamin B1 and emerin expression, nor affect the expression of the pluripotency markers Nanog, Oct-4 and Sox-2 (*Figure 2C*). TERT is expressed in undifferentiated ESC, but is repressed during differentiation (*Armstrong et al., 2000*). To determine whether ESC would become susceptible to progerin expression upon differentiation, we aggregated ESC into embryoid bodies (EB), plated them in tissue culture dishes and measured the size of the differentiating EB outgrowth upon plating. While the total EB size did not vary significantly between the different conditions (*Figure 2E*), the EB outgrowth of differentiated cells was significantly reduced in progerin expressing cells (*Figure 2F*, p < 0.01). As in primary fibroblasts, we observed an increase in 53BP-1 foci in the differentiated progerin expressing cells (*Figure 2—figure supplement 1C*). To further investigate whether TERT is necessary to prevent progerin-induced defects in pluripotent ESC, we expressed progerin in $Tert^{-/-}$ mouse ESC (*Figure 2—supplement 1D,E*). Expression of progerin in $Tert^{-/-}$ ESC led to a reduction in cell number (*Figure 2G*), rapidly induced differentiation and significantly impaired ability of ESC to form embryoid bodies (*Figure 2H,I*). Taken together, these results demonstrate that physiological expression levels of TERT are necessary and sufficient to prevent progerin-induced defects.

## BioID analysis reveals an impaired interaction between LAP2α and progerin

Cellular senescence is considered to be a key factor in HGPS, as well as during normal ageing in humans (*Kuilman et al., 2010*). To determine how progerin may trigger senescence, we compared the protein interactomes of lamin A and progerin using BioID (*Roux et al., 2012*). The Myc-tagged promiscuous biotin ligase BirA* was fused to the N-termini of lamin A or progerin, and expressed in fibroblasts by DOX-induction. To avoid complications from senescence-associated secondary consequences of progerin expression, we performed the comparison in TERT-expressing cells. Upon induction, BirA*-lamin A and BirA*-progerin were expressed (*Figure 3A*), localized at the nuclear

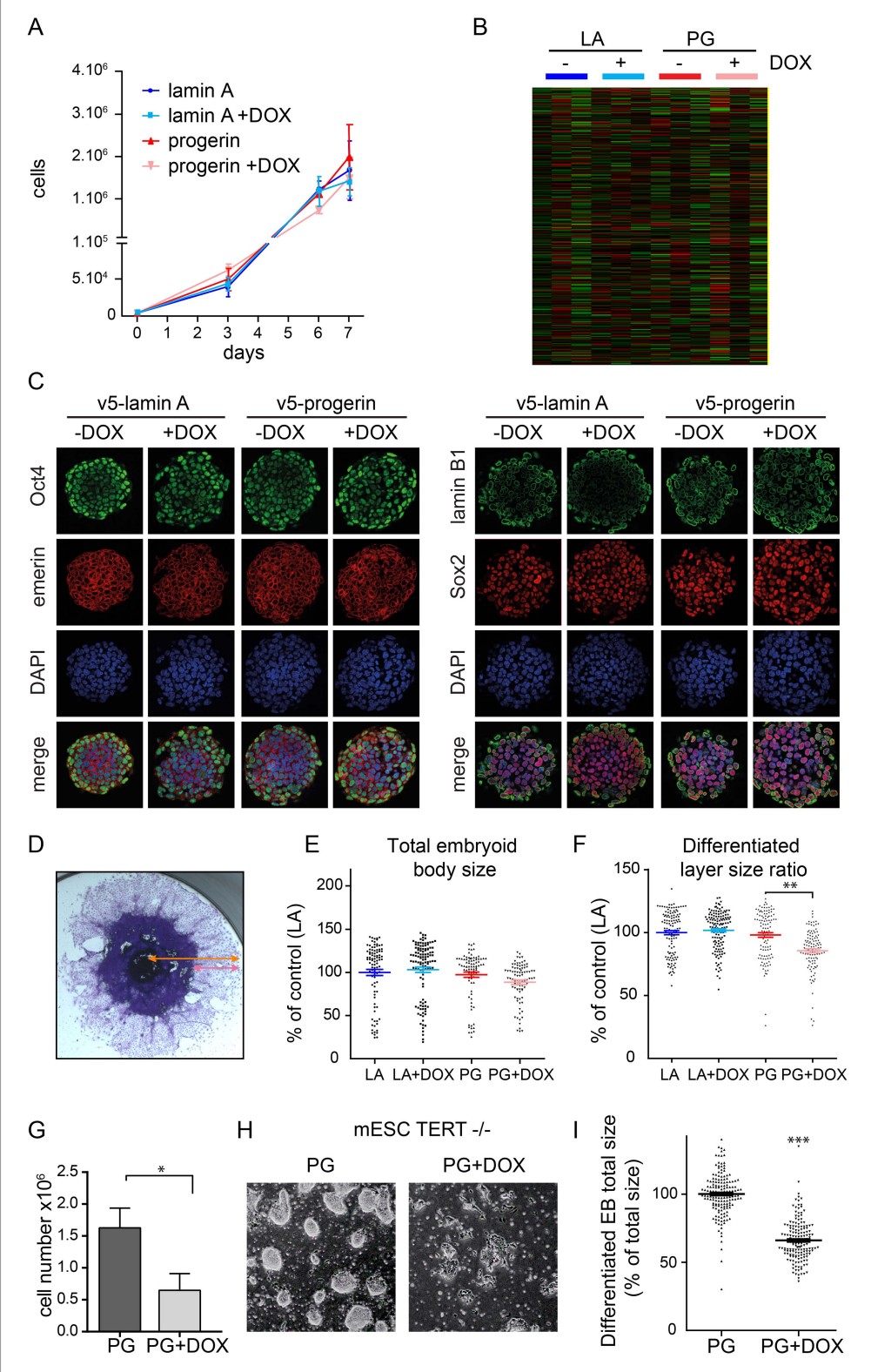

**Figure 2**. Physiological levels of telomerase prevent progerin-induced defects in mouse ESC. (**A**) Growth curve of mouse ESC expressing progerin (PG) or lamin A (LA) upon DOX induction (n = 3, error bars indicate SEM). (**B**) Heatmap showing the number of genes whose expression changed more than twofold after 8 days of lamin A or progerin expression (I, induced. N.I., non-induced). (**C**) Immunofluorescence microscopy using Oct-4, emerin, lamin

*Figure 2. continued on next page*

*Figure 2. Continued*

B1 and Sox2 antibodies in the presence or absence of v5-lamin A and v5-progerin expression. (**D**) Embryoid body (EB) formation upon removal of leukemia inhibitory factor (LIF). The orange line indicates the total size of the differentiated EB, while the pink line indicates the differentiated cell outgrowth. (**E**) Quantification of total embryoid body size in ESC expressing lamin A (LA+DOX) or progerin (PG+DOX), compared to EBs differentiated from ESC LA non induced controls (one-way ANOVA, n > 80, p > 0.05). (**F**) Quantification of the size of the differentiated cell layer, in percentage of the total EB size for each EB, compared to EBs differentiated from non-induced ESC LA controls (p < 0.01, n > 80, one-way ANOVA with Tukey's post-test). (**G**) Cell counts of *Tert*$^{-/-}$ ESC in the presence (PG+DOX) or absence (PG) of progerin. Cells were induced for 5 days prior to cell counting (p < 0.05, n = 3, Student's *t*-test). (**H**) Brightfield microscopy images of *Tert*$^{-/-}$ ESC ± progerin. Pictures were taken 7 days after induction with progerin (PG+DOX) or non-induced controls (PG). (**I**) Total size of EBs differentiated from *Tert*$^{-/-}$ ESC expressing progerin (PG+DOX) or controls (PG) (p < 0.001, n > 160, Student's *t*-test).

The following figure supplement is available for figure 2:

**Figure supplement 1**. (**A**) Western blot showing inducible expression of v5-progerin or v5-lamin A in primary mouse ESC ± doxycycline (DOX) as indicated.

periphery (*Figure 3B*), with BirA*-progerin inducing lobulated and misshapen nuclei (*Figure 3B*). Protein biotinylation by the BirA*-lamin A and progerin fusion proteins occurred exclusively upon addition of biotin and DOX (*Figure 3—figure supplement 1A*). Biotinylated proteins were purified and analyzed by mass spectrometry. As expected, self-biotinylated BirA*-lamin A, BirA*-progerin, endogenous lamin A/C and biotinylated lamin B1, previously shown to interact with A-type lamins, were identified (*Figure 3—figure supplement 1B,C*) (*Kubben et al., 2010*). Mass spectrometry analysis of pull-down fractions revealed several known components of the nuclear envelope/lamina, including lamin A, LAP2, emerin, lamin B1 and B2 (*Figure 3—figure supplement 1C*) (*Roux et al., 2012*). We compared the interactome of lamin A vs progerin, and quantified the differential interactions using the exponentially modified protein abundance index (emPAI) (*Ishihama et al., 2005*). We observed a decreased interaction of the nuclear pore complex protein TPR with progerin, consistent with a previous report describing impaired nuclear import of TPR in HGPS cells (*Snow et al., 2013*). A list of the 11 identified nuclear proteins and their respective interaction index with lamin A or progerin is shown in *Figure 3—figure supplement 1C*.

We observed a significantly decreased interaction of progerin with the lamina-associated polypeptide 2 (LAP2) (*Figure 3C*). LAP2 exists as several alternatively spliced isoforms (*Dorner et al., 2006*), among which LAP2α and β were identified by BioID. Since LAP2α forms nucleoplasmic complexes with lamin A (*Dechat et al., 2000*), and its levels decline with progerin expression (*Scaffidi and Misteli, 2005*; *Zhang et al., 2011*) or during senescence (*Dreesen et al., 2013*), we focused on the α-isoform. To avoid complications associated with cellular senescence, we used TERT+ cells expressing BirA*-progerin, in which total LAP2α levels remained stable in protein extracts (*Figure 3A*, lanes 3 + 4 bottom panel). This confirmed that the reduced interaction between LAP2α we observed by BioID was not due to a global decrease in the LAP2α levels in the protein samples.

In addition, we expressed BirA*-lamin A and BirA*-progerin in pluripotent ESC. Both constructs correctly localized to the nuclear periphery and did not lead to any alterations in the nuclear lamina, as judged by emerin, lamin B1 and SUN1 staining (*Figure 3—figure supplement 2A,B*). As expected, mass spectrometry analysis of pull-down fractions identified the nuclear lamina constituents lamin A/C, lamin B1 and B2, suggesting that the BirA*-constructs interact with endogenous proteins similarly in ESC and in human fibroblasts. We also noted that, progerin also showed a significantly decreased interaction with LAP2 in the ESC (*Figure 3—figure supplement 2C*).

To determine whether LAP2α physically interacts with lamin A and progerin, we examined the interaction of in vitro transcribed/translated v5-tagged lamin A or v5-progerin with LAP2α and emerin by co-translation followed by co-immunoprecipitation (*Figure 3D*, upper panel). Progerin consistently pulled down ∼40–60% less LAP2α than lamin A, while its interaction with emerin was unaffected (*Figure 3D*, lower panel). These results demonstrate that the weakened binding between progerin and LAP2α, suggested by the BioID screen, is due to a reduction of the association between progerin and LAP2α.

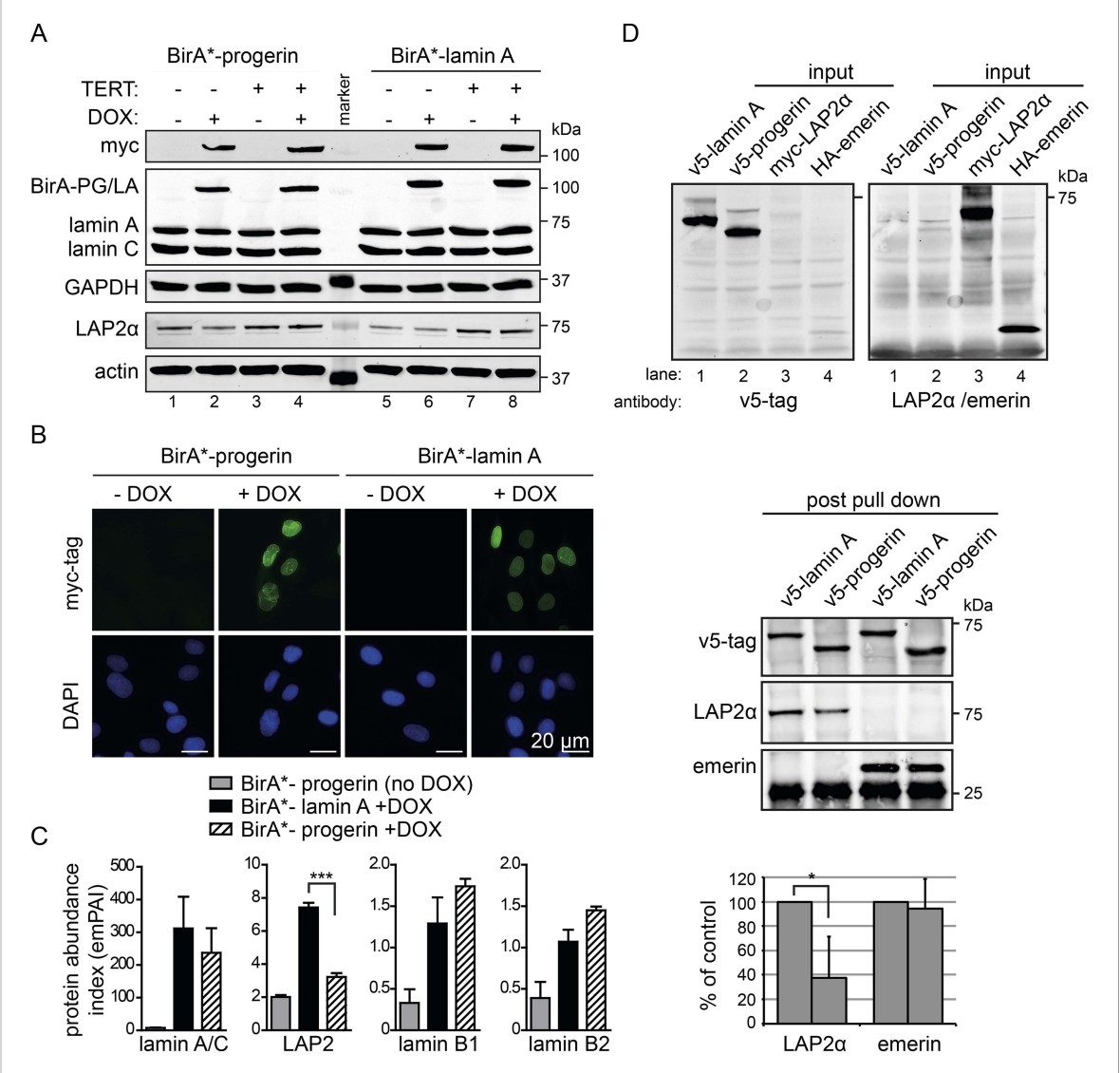

**Figure 3**. BioID analysis reveals differential interaction of lamin A and progerin with lamina-associated polypeptide 2 (LAP2). (**A**) Western blot showing doxycycline-dependent expression of myc-BirA*-progerin (BirA-PG) and myc-BirA*-lamin A (BirA-LA) fusion constructs in primary and TERT+ cells. Antibodies are indicated: myc; lamin A, lamin C, LAP2α, actin, GAPDH. (**B**) Immunofluorescence microscopy confirms doxycycline-dependent induction and localization of BirA*-lamin A/BirA*-progerin fusion constructs to the nuclear periphery (green, myc tag; blue, DAPI staining). Scale bar: 20 μm. (**C**) Impaired interaction of LAP2 with progerin. Quantitative interactome of lamin A (black bars) or progerin (striped bars) with nuclear proteins lamin A, LAP2, lamin B1 and B2. Control: non-induced BirA*-lamin A (grey bars). BioID (emPAI) index: quantification based on the number of peptides for each protein detected by mass spectrometry error bars represent SEM (n = 3, ***p < 0.001, one-way ANOVA with Tukey's post-test). (**D**) Interaction of lamin A or progerin with LAP2α or emerin by co-immunoprecipitation. In vitro transcribed and translated v5-tagged lamin A, v5-tagged progerin, LAP2α and emerin (antibodies: v5-tag, LAP2α, emerin are indicated). Top panel: recombinant v5-tagged progerin and lamin A, myc-LAP2α and HA-emerin were efficiently immunoprecipitated using anti-v5-tag or anti-myc antibodies, respectively (input lanes two, three and four). Bottom panel: LAP2α or emerin immunoprecipitated by either v5-lamin A or v5-progerin. Quantification of LAP2α and emerin pulled down by v5-lamin A or v5-progerin is shown below (normalized to respective v5-signal, *p < 0.05, Student's t-test).

The following figure supplements are available for figure 3:

**Figure supplement 1**. BirA*-dependent biotinylation of proteins in human fibroblasts, control experiments and protein list.

**Figure supplement 2**. BioID analysis of lamin A or progerin in pluripotent ESC.

## Super-resolution microscopy reveals impaired localization of LAP2α to telomeres in HGPS cells

A comparison of the lamin A and progerin interactomes has been described using other procedures (*Kubben et al., 2010*; *Liu et al., 2011*), but it is unclear whether any of the previously identified differential interactors had any functional role in the pathophysiology of HGPS. However, a previous report indicated that LAP2α may directly interact with chromatin and telomeres (*Dechat et al., 2004*). In addition, since exogenous telomerase suppresses progerin-induced defects, we investigated whether LAP2α localization to the telomeres was altered in TERT+HGPS cells. To address this with sufficient resolution, we used 3D-structured illumination microscopy (*Schermelleh et al., 2010*) to compare nuclei from TERT+ wild-type and TERT+HGPS cells. In normal nuclei, LAP2α was present as discrete foci distributed throughout the nucleoplasm (*Figure 4A,B*), with many foci closely localized with telomeres, visualized by staining for TRF1, a component of the telomere-associated shelterin complex (*De Lange, 2005*). We then measured the distribution profile of LAP2α along a ~400 nm axis

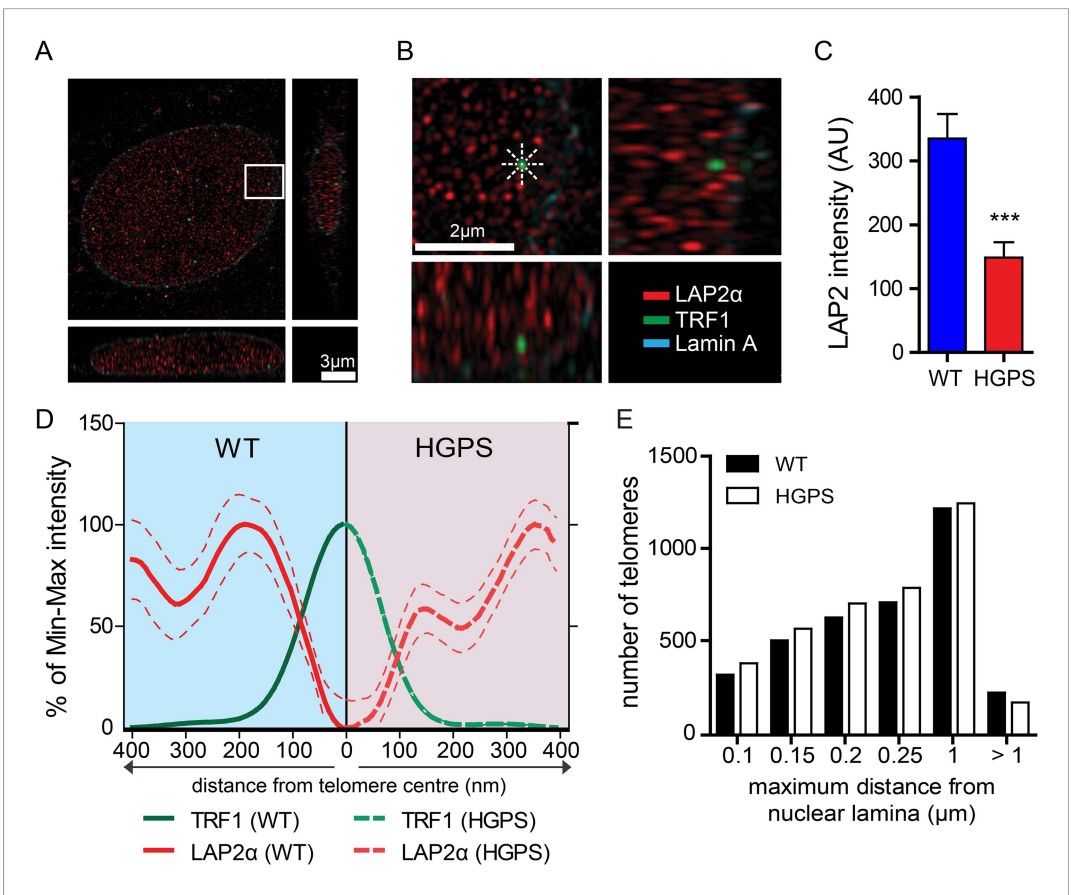

**Figure 4**. LAP2α association with telomeres. (**A**, **B**) Projection of an extended section of a wild-type fibroblast nucleus showing lamin A (blue), LAP2α (red) and TRF1 (green) staining. Magnified section indicated by the white frame is shown in panel (**B**). The eight radiuses used to measure the distribution profile of LAP2α around each telomere are indicated by the dotted white lines. (**C**) Average LAP2α intensity in WT or TERT+HGPS nucleus (n = 17, \*\*\*p < 0.001, errors bars indicate SEM, Student's *t*-test). (**D**) Intensity profile of LAP2α (red) near telomeres (TRF1, green) in WT (left part, blue) or TERT+HGPS (right part, red) nuclei along ~400 nm axis, relative to the minimum and maximum LAP2α and TRF1 signal per nucleus. Dotted lines indicate SEM, n ≥ 88. (**E**) Quantification of telomere distance to the nuclear lamina (a total number of 2891 telomeres from 34 nuclei were analyzed).

The following figure supplement is available for figure 4:

**Figure supplement 1**. LAP2α association with telomeres, control experiments and details of analysis.

from the center of each telomere in 3D-SIM nuclear sections (544 nm thick), at 45° angle intervals (*Figure 4B*). To eliminate any localization bias due to the observed differences in LAP2α amounts between WT and HGPS nuclei (*Figure 4C*, p < 0.001, n = 17), we normalized the signal intensity to the maximum/minimum LAP2α intensity for each nucleus. We found that in normal nuclei, the highest average LAP2α signal intensity is within ~200 nm of each telomere. In contrast, in HGPS nuclei (*Figure 4—figure supplement 1A,B*), the distribution of LAP2α in relation to the telomeres was significantly altered, and reached its maximum value roughly 360–400 nm from the center of each telomere (*Figure 4D*). We observed a similar profile of LAP2α localization to telomeres in TERT+ wild type cells expressing progerin (*Figure 4—figure supplement 1H,I*).

To confirm this loss of proximity between LAP2α and the telomeres in HGPS cells, we quantified the surface area of telomeres co-localizing with LAP2α for each telomere in normal or HGPS nuclear sections. Consistent with our previous results, we found a decrease in the surface area of HGPS nuclei associated with LAP2α (6.7% of the surface of each HGPS nuclei was covered by LAP2α vs 15.7% for wild-type nuclei [p < 0.001, n = 25, *Figure 4—figure supplement 1C*]). To take into account this difference in nucleoplasmic LAP2α levels between HGPS and wild type nuclei, we grouped telomeres into three categories (i) low co-localization TRF1/LAP2α at the telomeres: telomeres with a percentage of their surface co-localizing with LAP2α below the average percentage of surface covered by LAP2α in the nucleus, minus one standard deviation (noted $< \overline{LAP2c} - \sigma$), (ii) average co-localization TRF1/LAP2α at the telomeres: telomeres with a percentage of their surface co-localizing with LAP2α within the average surface covered by LAP2α in the nucleus (noted $\overline{LAP2c} \pm \sigma$) and (iii) high co-localization TRF1/LAP2α at the telomeres: telomeres with a percentage of their surface co-localizing with LAP2α above the average surface area covered by LAP2α in the nucleus plus one standard deviation (noted $> \overline{LAP2c} + \sigma$). Examples of these quantified images used for telomere/LAP2α co-localization are shown in *Figure 4—figure supplement 1D,E*. In agreement with an impaired LAP2α localization at telomeres, we observed a lower co-localization of telomeres with LAP2α in HGPS cells. In addition, we observed fewer telomeres with average or high LAP2α association in HGPS cells as indicated by categories $\overline{LAP2c} \pm \sigma$ and $> \overline{LAP2c} + \sigma$, respectively (*Figure 4—figure supplement 1F*). Taken together, these results demonstrate that the close physical proximity between LAP2α and telomeres is disrupted in HGPS nuclei.

## Although telomeres localize at the nuclear periphery, their localization is not significantly affected by progerin

Depletion of, or mutations in *LMNA* can alter telomere distribution within the nucleus (*Gonzalez-Suarez et al., 2009*; *Taimen et al., 2009*; *De Vos et al., 2010*). Telomeres may also transiently localize to the nuclear periphery during the G1 phase of the cell cycle (*Crabbe et al., 2012*). To determine whether telomeres were mis-localized in HGPS nuclei, we used 3D-SIM and 3D rendering to measure the distance between telomeres and the nuclear lamina (*Figure 4—figure supplement 1G*). We found that ~50% of the telomeres localized to within 250 nm of the nuclear lamina in interphase nuclei, but we did not observe any change in telomere distribution between normal- and HGPS+TERT fibroblasts (*Figure 4E*). This suggests that progerin expression or mis-localization of LAP2α does not affect telomere distribution within the nucleus.

## LAP2 deficiency exacerbates progerin-induced proliferation defects

To investigate whether the impaired association of LAP2α with telomeres and progerin was functionally relevant to the pathophysiology of HGPS, we modulated LAP2 levels in wild type and progerin expressing cells using our doxycyclin-inducible system. First, we depleted the α, β and γ isoforms of LAP2 using lentiviral delivered shRNA, and observed, in agreement with previous reports, enhanced proliferation of primary and TERT+ fibroblasts (*Figure 5—figure supplement 1A–C*) (*Dorner et al., 2006*; *Naetar et al., 2008*). To determine the consequences of LAP2 depletion in progerin expressing cells, we introduced v5-tagged progerin or lamin A into LAP2-depleted fibroblasts (*Figure 5B*). Surprisingly, and in contrast to normal cells, the loss of LAP2 enhanced progerin-induced proliferation defects (*Figure 5C*, red arrowhead). However, this enhanced reduction in proliferation was rescued by TERT expression (*Figure 5—figure supplement 1D*). From these findings we conclude that LAP2 depletion potentiates the detrimental effect of progerin on cell proliferation.

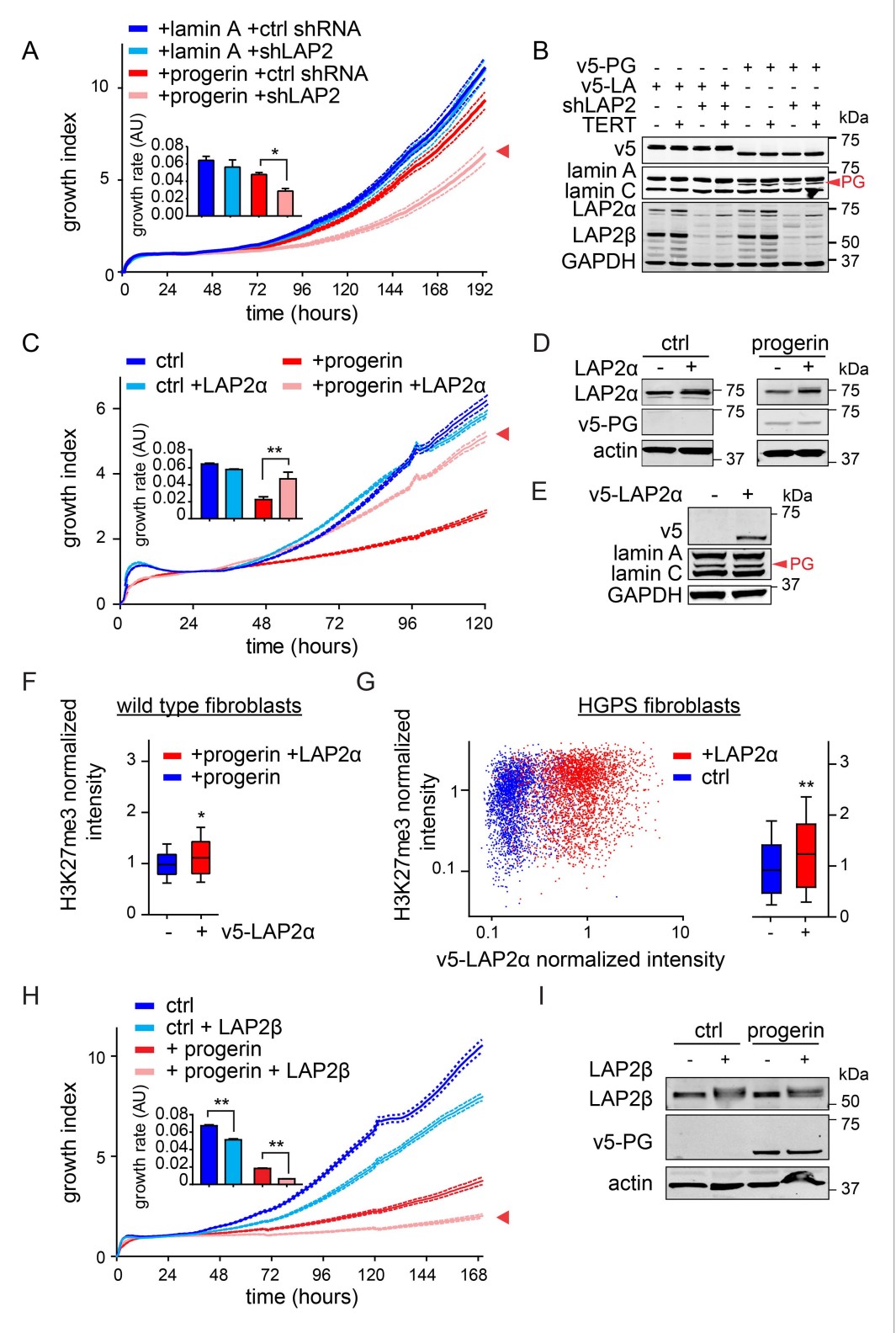

**Figure 5**. LAP2 depletion exacerbates the progerin-induced proliferation defect whereas specific overexpression of LAP2α rescues it. (**A**) Growth curve of normal (ctrl shRNA) and LAP2-depleted (shLAP2) primary fibroblasts expressing progerin or lamin A. Dotted lines indicate SEM (n = 3). Inset: growth rate after 8 days, error bras indicate SEM (*p < 0.05, Student's *t*-test). (**B**) Western blot of primary and TERT+ cells expressing v5-tagged lamin A (v5-LA)

*Figure 5. continued on next page*

*Figure 5. Continued*

or progerin (v5-PG) with or without LAP2 silencing by shRNA (shLAP2). V5-tag, lamin A, lamin C, progerin (PG), LAP2α, LAP2β and GAPDH are indicated. (**C**) Growth curve of fibroblasts expressing control vector or progerin in the presence or absence of doxycycline-inducible LAP2α. LAP2α expression was induced by addition of 0.25 µg/ml doxycycline. Dotted lines indicate SEM (n = 3). Inset: growth rate after 5 days, error bras indicate SEM (**p < 0.01, Student's *t*-test). (**D**) Western blot of primary fibroblasts carrying doxycycline-inducible v5-LAP2α, expressing control vector or progerin. LAP2α expression was induced by addition of 0.25 µg/ml doxycycline. LAP2α, progerin and actin are indicated. (**E**) Western blot showing doxycycline-dependent induction of v5-LAP2α in TERT+HGPS cells. (**F**) Box plot of H3K27me3 levels in human fibroblasts expressing progerin in the presence (red) or absence (blue) of ectopically expressed LAP2α (Student's *t*-test, p < 0.05, n > 7500 cell analyzed, whiskers represent 10–90 percentile. (**G**) Scatter plot analysis of H3K27me3 levels in TERT+HGPS cells in the presence (red) or absence (blue) of ectopically expressed LAP2α showing increased levels of H3K27me3 upon LAP2α induction (Student's *t*-test, p < 0.01, n > 4000 cell analyzed. Inset: box plot of the same data, whiskers represent 10–90 percentile, ***p < 0.001). (**H**) Growth curve of control (ctrl) or progerin expressing fibroblasts (+progerin) in the presence or absence of doxycycline-inducible v5-LAP2β. Dotted lines indicate SEM (n = 3). Inset: growth rate after 6 days, error bars indicate SEM (**p < 0.01, Student's *t*-test). (**I**) Western blot of fibroblasts transduced with doxycycline-inducible LAP2β and expressing control vector or progerin. LAP2β, progerin and actin are shown.

The following figure supplements are available for figure 5:

**Figure supplement 1**. Depletion and expression of LAP2 in wild type and progerin expressing fibroblasts, control experiments.

**Figure supplement 2**. Expression of LAP2α prevents progerin induced DNA damage and premature senescence.

**Figure supplement 3**. Effects of valproic acid treatment on proliferation of control (non-induced WT, green lines) and progerin-expressing (red lines) normal dermal fibroblasts.

## Increased LAP2α specifically rescues progerin-induced defects

Since depletion of LAP2 enhances the progerin-induced reduction in proliferation, we investigated whether ectopic expression of specific LAP2 isoforms would ameliorate progerin-induced defects. In normal cells, increased expression of LAP2α impairs cell proliferation in a dose-dependent manner (*Figure 5—figure supplement 1E–G*), as previously reported (*Dorner et al., 2006*). We then introduced progerin into cells expressing exogenous LAP2α (*Figure 5C,D*). Strikingly, a moderate increase in LAP2α levels almost completely restored the proliferative capacity of cells expressing progerin (*Figure 5C*, red arrowhead), reduced the number of cells with DNA damage foci (*Figure 5—figure supplement 2A*) and prevented premature senescence in these cells (*Figure 5—figure supplement 2B,C*). This is in contrast to wild-type cells (or vector control cells), in which increased LAP2α impaired proliferation (*Figure 5—figure supplement 1E–G*) and increased premature senescence (*Figure 5—figure supplement 2B*).

To determine to which extent LAP2α prevented progerin-induced defects, we used immunofluorescence microscopy to measure LAP2α and H3K27me3 levels in single cells upon LAP2α induction. As shown in *Figure 5F*, expression of LAP2α in progerin expressing cells increased H3K27me3 levels as compared to non-induced controls (n = 3, p < 0.05). We further confirmed these results in TERT+HGPS fibroblasts ectopically expressing LAP2α (*Figure 5E,G*). Scatterplot analysis of ≈5500 nuclei revealed a significant increase in H3K27me3 upon induction of LAP2α (Pearson r = 0.1835, p < 0.01) (n = 3, p < 0.001). However, overexpression of LAP2α did not ameliorate progerin induced nuclear abnormalities in TERT+HGPS fibroblasts (*Figure 1—figure supplement 2B*). Lastly, overexpression of LAP2α in wild type cells, not expressing progerin, did not result in a significant increase in H3K27me3 levels (*Figure 5—figure supplement 1H*). Taken together, these results suggest that increasing LAP2α levels prevents progerin-induced proliferation defects and alleviates the progerin-induced reduction in the heterochromatin mark H3K27me3.

A reduction in heterochromatin renders DNA vulnerable to increased damage (*Di Micco et al., 2011*). To determine whether 'open chromatin' renders cells more susceptible to progerin, we treated progerin-expressing, and their non-induced (wild type) controls, to increasing concentrations of the histone deacetylase inhibitor valproic acid (0.025–0.5 µM). As shown in *Figure 5—figure supplement 2*,

valproic acid treatment exacerbated progerin-induced proliferation defects, at concentrations that had no discernible effect on non-induced wild type cells (*Figure 5—figure supplement 3*). Based on these results we speculate that a reduction in heterochromatin may render cells more susceptible to progerin-induced proliferation defects.

LAP2 exists in many isoforms. To determine whether the rescue by LAP2α is specific to this isoform, we expressed the β-isoform of LAP2 in normal- and progerin expressing cells (*Figure 5H,I*; *Figure 5—figure supplement 1I–K*). In contrast to LAP2α, ectopic expression of LAP2β led to a reduction in the rate of proliferation in both normal and progerin expressing fibroblasts (*Figure 5H*, red arrow). Together, these results demonstrate that expression of the α-isoform of LAP2 specifically ameliorates progerin-induced proliferation defects and increases the levels of heterochromatin associated H3K27me3.

## Discussion

HGPS is described as a 'segmental ageing syndrome', but it remains unclear why specific tissues are more affected, in particular those of mesenchymal origin, while others, such as neural lineages, are seemingly spared (*Zhang et al., 2011*). To address this, we developed a DOX-inducible expression system to regulate the levels and timing of progerin expression in isogenic primary and TERT-positive cells. By using this system, we find that progerin inhibits proliferation, causes DNA damage and entry into senescence in a dose-dependent fashion. These results suggest that progerin's detrimental effects become apparent only when levels reach a critical threshold. This provides a compelling explanation as to why tissues expressing relatively high levels of lamin A/progerin are central to the pathology of HGPS (*Jung et al., 2012*; *Nissan et al., 2012*).

Previous studies showed that progerin-induced defects can be rescued by ectopic expression of TERT (*Kudlow et al., 2008*; *Benson et al., 2010*). However, it remained unclear whether physiological levels of TERT would suffice, and to what extent TERT prevents progerin's defects. Our results confirm that ectopic expression of TERT prevents progerin-induced DNA damage, proliferation defects, premature senescence and senescence-associated loss of lamin B1 (*Kudlow et al., 2008*; *Benson et al., 2010*). Moreover, by expressing progerin and lamin A in ESCs which express endogenous levels of TERT, and *Tert*⁻ᐟ⁻ ESC, we demonstrate that physiological levels of TERT are sufficient to prevent progerin-induced proliferation defects and changes in gene expression. These results may be relevant to HGPS patients, as TERT expression during embryogenesis or in adult stem cell compartments may protect these cells from the detrimental consequences of progerin (*Wright et al., 1996*). In this respect, HGPS may be quite different from dyskeratosis congenita (DC), a premature ageing disorder that is caused by defects in telomerase that particularly affects stem cell maintenance (*Vulliamy et al., 2004*).

Here we have expanded the analysis as to what physiological and biochemical parameters are effected by progerin, and to what extent they are restored to normal levels by the simultaneous expression of TERT. Importantly, the microarray analysis revealed that the persistent loss of H3K27me3 in TERT-positive progerin expressing cells did not result in significant changes in gene expression. However in determining the levels of the heterochromatin marker H3K27me3, in primary and TERT-positive cells, we found that TERT does not prevent progerin-induced loss of heterochromatin. This, in turn, suggests that the changes in gene expression in primary fibroblasts expressing progerin are a consequence of premature senescence, rather than a direct consequence of progerin expression.

To determine the direct effects of progerin on other nuclear proteins, with the consequent impairment of cell proliferation and premature senescence, we compared the protein interactomes of lamin A and progerin. Previous studies showed that lamin A interacts with LAP2α (*Dechat et al., 2000*; *Kubben et al., 2010*). Our interactome analysis confirmed this interaction, but demonstrated that the physical interaction between LAP2α and progerin was significantly reduced. Furthermore, it had been suggested that LAP2α may transiently associate with telomeres during mitosis (*Dechat et al., 2004*), and cells with a disrupted nuclear lamina show abnormal telomere localization (*Gonzalez-Suarez et al., 2009*; *Taimen et al., 2009*). We therefore compared telomere distribution, and their association with LAP2α, in HGPS and normal nuclei. We found that a significant proportion of telomeres can be found at the nuclear periphery, and that their distribution is not perturbed by progerin. This suggests that other progeroid mutations (*LMNA* E145K), or loss of *LMNA* may have different effects on telomere localization (*Gonzalez-Suarez et al., 2009*; *Taimen et al., 2009*).

Although telomere distribution was unaffected, we find that the association of LAP2α with telomeres is reduced in HGPS.

To determine the functional relevance of these changes, we modulated LAP2α levels in normal and progerin-expressing fibroblasts. Depletion of LAP2 exacerbated the progerin-induced defects whereas increasing nuclear LAP2α levels prevented the proliferation defects, DNA damage and premature senescence resulting from progerin expression. This rescue was specific to the α-isoform of LAP2, as increasing LAP2β levels failed to restore normal rates of proliferation. Moreover, increased or decreased levels of LAP2α led to diametrically opposed effects between normal and progerin-expressing cells. Taken together, our data provide mechanistic evidence that LAP2α plays a key role in the HGPS pathophysiology and that progerin-induced defects are rescued by ectopic expression of LAP2α.

Changes in the levels of LAP2α are emerging as a consistent feature in at least some of the laminopathies. Loss of *Lmna*, that causes cardiomyopathy and muscular wasting/dystrophy results in a transient increase in LAP2α in muscle precursors and myoblasts (*Melcon et al., 2006*). However, ablation of LAP2α in *Lmna* null mice significantly reduces the severity of the disease and increases longevity (*Cohen et al., 2013*), a result similar to that observed following elimination of another nuclear envelope protein Sun1 (*Chen et al., 2012*). Together these observations, and those presented here, suggest that the lamina/nuclear envelope forms an integrated and mutually regulated stoichiometric network of proteins. When the network is disrupted, for example by *LMNA* mutations, one of the consequences is that the levels of other protein components change, with this change being a major contributing factor to the consequent pathology.

Within this context, and based on the results presented here, we propose a model linking progerin and LAP2α to human telomeres (*Figure 6*). In this model, telomeres in normal cells are surrounded by LAP2α complexes, which in turn interact with nuclear lamins (*Figure 6A*). In progeria, the impaired association between LAP2α and progerin disrupts this organization and ultimately leads to premature senescence (*Figure 6B*). This can be prevented by expression of either telomerase or by increasing the levels of LAP2α (*Figure 6C,D*). However, it is not clear if the effect of TERT mechanistically differs from the rescue by LAP2α. Our data suggest that this indeed might be the case: in contrast to LAP2α, TERT expression does not ameliorate the progerin-induced reduction in H3K27me3 levels. However, it remains to be investigated whether progerin affects H3K27me3 specifically at telomeric- or subtelomeric regions, and how expression of LAP2α ameliorates this loss. Telomeres are fragile sites and a reduction in heterochromatin has been associated with increased susceptibility to DNA damage and cell cycle arrest (*Sfeir et al., 2009*; *Di Micco et al., 2011*). In agreement with this hypothesis, progerin expression destabilizes chromosome ends, causing DNA damage and premature senescence (*Benson et al., 2010*; *Wood et al., 2014*). By elongating telomeres, TERT counteracts such telomere loss and prevents the telomeric DNA damage associated with HGPS (*Figure 6E*; *Figure 1—figure supplement 1I,J*) (*Kudlow et al., 2008*; *Benson et al., 2010*). In contrast to TERT, we found that increased LAP2α alleviates the progerin-induced loss of H3K27me3 (*Figure 5F,G*). Chromatin decondensation by treatment with valproic acid enhanced progerin-dependent pro-liferation defects. Taken together, our results suggest that increased LAP2α stabilizes chromatin structure by increasing H3K27me3 and prevents progerin-associated DNA damage that resulted in premature senescence (*Figure 6E*). In support of this notion, both LAP2α and H3K27me3 decline during normal ageing in human cells (*Scaffidi and Misteli, 2006*; *Naetar et al., 2007*; *Dreesen et al., 2013*), and it has been suggested that sustaining heterochromatin levels may extend lifespan by protecting against DNA damage (*Larson et al., 2012*). Our results have important implications in understanding the role of the nuclear architecture, in particular the lamina and nuclear envelope, in regulating cell proliferation, chromatin organization and in providing novel insights into the molecular pathology of progeria. They may also be relevant to other laminopathies that are associated with perturbations or mutations of components of the nuclear lamina.

## Materials and methods

### Fibroblast cell culture, immunofluorescence and image acquisition

Normal primary dermal fibroblasts were a gift from Dr Bruno Reversade (Institute of Medical Biology, A*STAR, Singapore). Fibroblasts were grown under standard culture conditions (37˚C; 5% CO$_2$) in minimum essential medium (MEM; Invitrogen, Carlsbad, CA) supplemented with 50 U/ml penicillin and streptomycin (Invitrogen), 15% fetal calf serum (FBS, Invitrogen), 0.2 mM non-essential amino

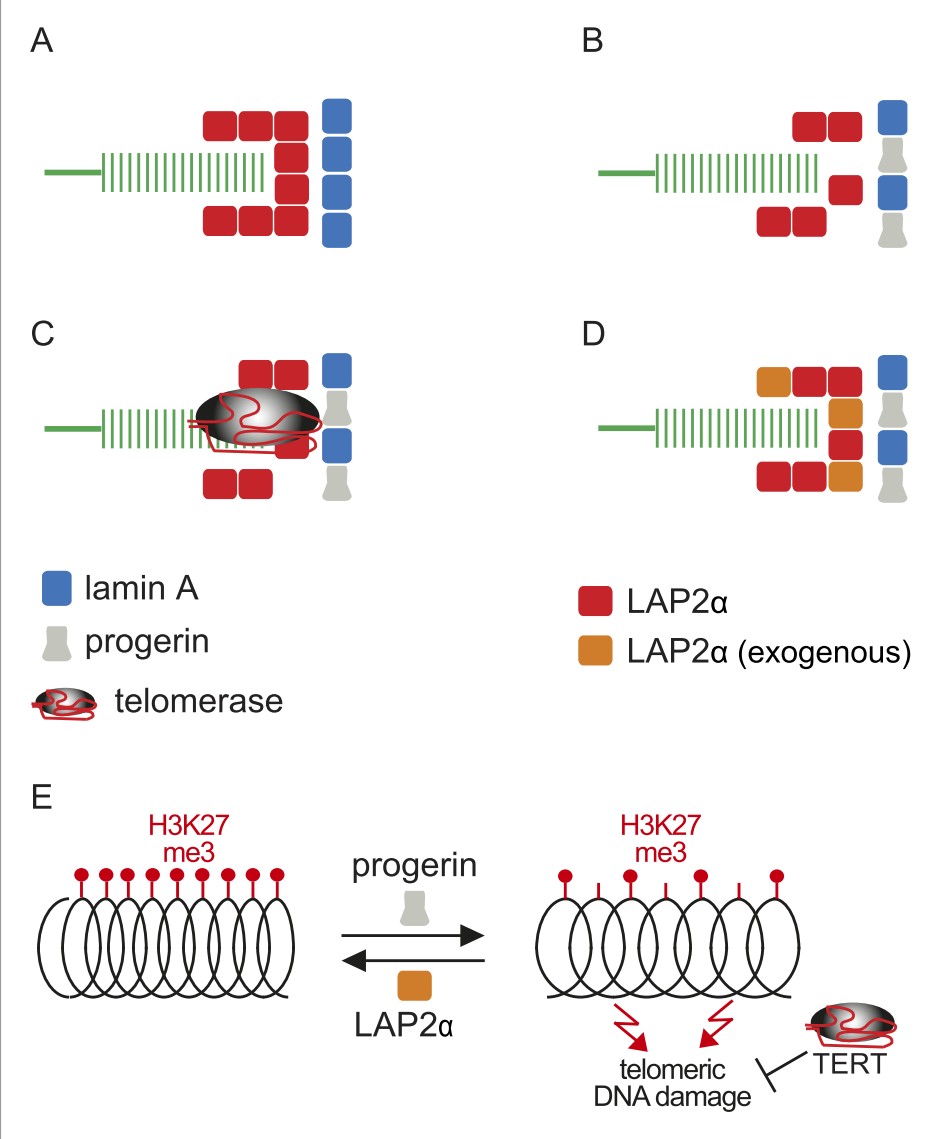

**Figure 6**. Putative model of lamin A/progerin::LAP2α interaction mechanism with telomeres. (**A**) Normal formation of lamin A::LAP2 complexes allows proper positioning of LAP2α near telomeres (green). (**B**) Perturbed progerin:: LAP2α interaction impairs LAP2α localization at telomeres. (**C**) Telomeric damage resulting from this impaired interaction can be rescued by expression of telomerase (gray) or (**D**) by supplementing cells with exogenous LAP2α (orange). (**E**) Progerin-induced H3K27me3 loss is prevented by ectopic LAP2α expression. TERT expression does not prevent progerin-induced loss of H3K27me3 but rescues telomere dysfunction by telomere elongation.

acids (NEAA, Invitrogen) and 2 mM glutamine (Invitrogen). For immunofluorescence microscopic analysis, fibroblasts were grown on Lab Tek Chambered Coverglass slides (Nalge Nunc International, Rochester, NY) for 4–7 days, fixed 10 min in 4% paraformaldehyde, washed in PBS, permeabilized using PBS + 0.3% Triton-X and blocked in PBS + 5% FBS + 1% BSA. Primary antibodies in blocking solution were incubated overnight at 4°C, washed in PBS, probed with secondary antibodies for 30–45 min at room temperature (RT) and DAPI (4,6 diamidino-2-phenylindole) stained in PBS for 5 min. Images were acquired using a Zeiss Axiovert 200M inverted microscope (Carl Zeiss International, Singapore) using 10× NA 0.3 ZEISS Plan-NeoFluar, 40× NA 0.60 Ph2 Korr LD-Plan-NeoFluar or 63× NA 1.4 oil DIC ZEISS Plan-Apochromat objectives and a AxioCam MRm. Images were processed and exported using AxioVision LE software 4.5 SP (2006). Images were cropped and figures assembled using Adobe Photoshop CS4 and Adobe Illustrator CS3. DNA damage (by 53BP and γH2A-X staining)

was quantified by scoring 350–500 cells for each cell line and condition. Confocal images were acquired on an upright Olympus FV-1000 confocal microscope using a 100× oil objective.

## Constructs, retro- and lentiviral production, infection and selection

pBABE-Neo-hTERT (Counter et al., 1998; Hahn et al., 1999) were obtained from Addgene. Full length lamin A, progerin and LAP2β were amplified from a cDNA library (human embryonic stem cell line H9). LAP2α was amplified from pTD15 (gift from Dr Roland Foisner, Vienna). cDNAs were cloned into retroviral vector pBABE-hygro (Addgene, Cambridge, MA) or doxycyclin-inducible lentiviral vector pTRIPZ (Open Biosystem, Singapore). Restriction sites and v5-tag were introduced during PCR amplification step. Retroviruses were generated and fibroblast cultures infected using standard procedures. Lentiviruses were generated according to manufacturer's protocol (OpenBiosystem). Doxycline-dependent expression was verified by western blotting, immunofluorescence and FACS analysis.

## Antibodies

DNA damage foci were detected using antibodies against 53BP1 (Novus Biologicals; NB100-304) and anti-phospho-Histone H2A-X (Ser139) (Millipore; 05-636), lamin B1 (YenZym), lamin A/C (Millipore; MAB3211), progerin (Santa Cruz, SC 81611), LAP2 (Santa Cruz, H-130), LAP2α (Abcam, Ab 5162), TRF1 (Abcam 10579), V5-tag (Invitrogen; 37-7500), myc (Santa Cruz, sc-40), GAPDH (Sigma; G9545), β-tubulin (Covance; MRB 435P), β-actin (Sigma; A5441).

## Proliferation assays

DOX-induced and non-induced cells were seeded in triplicates, grown for 3–5 days, trypsinized and counted using a Scepter cell counter (Millipore). Experiments were repeated 2, 3 and 4 weeks after doxycycline induction. Growth curves were performed at least in triplicates using the xCELLigence System (Roche, Basel, Switzerland). Cell Index was monitored at hourly intervals.

## Mouse embryonic stem cell derivation, culture and differentiation

Tert$^{-/-}$ embryonic stem cells were derived by crossing heterozygous TERT-deficient mice (Jackson Laboratories; B6.129S-Tert$^{tm1Yjc}$/J). Embryonic stem cells were isolated from day 4 blastocyst stage embryos according to previously published protocols (Wong et al., 2010) and allowed to hatch out for 5 days. Outgrowths were dissociated and embryonic stem cells were expanded in KO-DMEM media supplemented with leukemia inhibitory factor (LIF), under standard culture conditions (37°C; 5% $CO_2$). Two wild-type, two homozygous and four heterozygous mouse ESC lines were genotyped using primers and conditions provided on the Jackson Lab website (http://jaxmice.jax.org/strain/005423.html). Bruce4 mouse embryonic stem cells (derived from C57BL/6 mouse strain) were grown under standard culture conditions (37°C; 5% $CO_2$) in 90% Knockout DMEM high glucose medium (Gibco, Waltham, MA), supplemented with 10% FBS, L-glutamine (2 mM, Gibco), pen strep (100 U/ml, Gibco), mercapto-ethanol (100 µM, Gibco), human LIF (10 ng/ml, Millipore), BIO GSK3-I (2 µM final, Calbiochem, Singapore). Differentiation was induced by removing leukemia inhibitory factor (LIF) from the culture medium as previously described (Keller, 1995). Embryoid bodies were generated using the hanging drop method, by aggregating 400 cells in 20 µl drops as described previously (Dang et al., 2002). To induce differentiation, embryoid bodies were grown for 6 days prior to plating onto gelatin coated dish in standard ESC growth medium without LIF. This study was performed in strict accordance with the recommendations in the Guide for the Care and Use of Laboratory Animals of the National Institutes of Health. All of the animals were handled according to approved institutional animal care and use committee (IACUC) protocols (140960) of the Institute of Medical Biology, A*STAR, Singapore.

## Immunoblotting

Whole cell lysates were isolated using Complete Lysis-M solution kit (Roche), quantified using the Pierce Microplate BCA protein assay kit (Thermo Scientific, Waltham, MA), separated by SDS-PAGE and transferred onto nitrocellulose membranes. Membranes were blocked for 1 hr in Odyssey Blocking Buffer:PBS (1:1) (LI COR Biosciences, Lincoln, NE) and hybridized with antibodies overnight at 4°C. Membranes were washed in PBS and two color detection was carried out using Odyssey Infrared (IR)-labeled secondary antibodies. A LI-COR Odyssey scanner was used to scan membranes and quantify signals.

## Microarray analysis

Primary and TERT+ fibroblasts stably expressing pTRIPZ-v5-lamin A or pTRIPZ-v5-progerin were grown in triplicates for 28 days in the presence or absence of 1 μg/ml doxycycline. At each time point, cells were grown to confluency and serum starved for 24 hr. Total mRNA was isolated using RNAeasy mini kit (Qiagen, Singapore) and integrity of the RNA was verified using Agilent 2100 Bioanalyzer (Agilent, Singapore). cRNA was synthesized using the Ambion Target Amp kit (Ambion) according to the manufacturer's protocol, and cRNA from each sample was hybridized to BeadChip v2 chips (Illumina).

## BirA*-fusion proteins, biotinylation and affinity capture of biotinylated proteins

Lamin A and progerin were N-terminally tagged with the myc-BirA* biotinylation enzyme and cloned into the pTRIPZ lentiviral vector. Primary and TERT+ fibroblasts stably expressing either construct were generated by lentiviral transduction and selected with 1.0 μg/ml puromycin. The myc-BirA*-fusion constructs were expressed upon induction with doxycycline for at least 6 days prior to analysis. Induction was verified by western blotting and by immunofluorescence microscopy using an anti-myc antibody (Santa Cruz, sc40). 50 μM biotin was added to the medium for 24 hr prior to lysing cells under denaturing conditions (M-lysis buffer; Roche). Control cells not induced with doxycycline, or without addition of biotin were processed in parallel. Biotinylated proteins were purified using streptavidin-coupled magnetic beads (Invitrogen). After reduction and alkhylation, purified proteins were separated by SDS-PAGE electrophoresis and analyzed by mass spectrometry. Proteins were quantified using the Exponentially Modified Protein Abundance Index (emPAI), which is directly proportional to the abundance of a protein in a mixture (*Ishihama et al., 2005*).

## In vitro translations and immunoprecipitations

V5-tagged lamin A, v5-tagged progerin and LAP2α cDNAs were cloned into pcDNA 3.1 (Invitrogen), and corresponding proteins were individually translated in vitro using the TnT quick coupled transcription/translation system (Promega) according to manufacturer's protocol. After translation, 20 μl of each produced protein was mixed with respective partners as indicated in the figure legend. 4 μg of anti-v5 antibody (Invitrogen) was added to each protein mix and incubated for 12 hr at 4°C with 50 μl of protein–G coupled Dynabeads (Invitrogen) in PBS. Beads were washed twice for 10 min in 0.5% sodium deoxycholate, 150 mM NaCl, 1% NP-40, 0.1% SDS, 50 mM TRIS pH7.4 with proteinase inhibitors, and once for 10 min in 20 mM TRIS HCl pH 7.4 with proteinase inhibitors. Protein complexes retained by the anti-v5 coupled beads were then eluted in Laemmli BioRad buffer at 95°C, run on SDS page gels and analyzed by western blotting using antibodies against v5-tag and LAP2α. Recovered amounts of LAP2α were quantified and normalized to the v5-tagged proteins (lamin A and progerin).

## Senescence-associated-β-gal staining

Cells were fixed in 2% formaldehyde and 0.2% glutaraldehyde for 5 min at room temperature, washed twice in PBS and incubated for 6 hr in 5-bromo-4-chloro-3-indolyl-β-D-galactopyranoside as described previously (*Dimri et al., 1995*).

## 3D-SIM super-resolution microscopy and image analysis

Cells were grown on microscopy cover glasses in 6-well plates, fixed in 2% paraformaldehyde in PBS for 20 min at room temperature and incubated in 50 mM NH4Cl/PBS (5 min) and 1% Triton X-100/0.1% SDS (5 min) according to a previously published protocol (*Dechat et al., 2004*). Samples were blocked in 0.2% gelatin/PBS for 30 min prior to antibody incubation. Primary and secondary antibodies were applied in gelatin/PBS for 1 hr at room temperature, washed in PBS and post-fixed in 2% paraformaldehyde in PBS for 20 min at room temperature.

Acquisition was performed using a DeltaVision OMX v4 Blaze microscope (GE Healthcare, Singapore), with the BGR-FR filter drawer for acquisition of 3D-SIM images. Olympus Plan Apochromat 100×/1.4 PSF oil immersion objective lens was used, with liquid-cooled Photometrics Evolve EM-CCD cameras for each channel. 15 images per section per channel were acquired with a z-spacing of 0.125 μm (*Gustafsson et al., 2008*; *Schermelleh et al., 2008*). Structured illumination reconstruction and wavelength alignment was completed using the SoftWorX (GE Healthcare)

program. 3D image rendering and analysis was performed using Imaris version 7.6 (Bitplane an Oxford Instruments Company), Tango (*Ollion et al., 2013*) and 2D image analysis using Fiji (*Schindelin et al., 2012*), and CellProfiler (*Carpenter et al., 2006*).

## Statistical analysis

Data and statistical analyses were performed using Excel and Graphpad Prism software. Results are shown as mean $\pm$ S.E.M/SD and box plots whiskers indicate 10–90 percentile, unless otherwise indicated. Data were analyzed using one or two way ANOVA and Bonferroni's/Tukey's post-hoc test if required, as well as two tailed Student's *t*-test and Pearson correlation coefficients, as appropriate. p-values below 0.05 were considered significant.

## Acknowledgements

We thank Ray Dunn, Leah Vardy and Nicolas Barker (Institute of Medical Biology, A*STAR Singapore) for discussions and reading the manuscript. We thank Vinay Tergaonkar for providing us with $Tert^{-/+}$ mice. This work was supported by the Singapore Biomedical Research Council and the Singapore Agency for Science, Technology and Research (A*STAR) and a grant from the Progeria Research Foundation to OD and CLS.

## Additional information

### Funding

| Funder | Author |
| --- | --- |
| Progeria Research Foundation (PRF) | Colin L Stewart, Oliver Dreesen |
| Agency for Science, Technology and Research (A*STAR) | Alexandre Chojnowski, Peh Fern Ong, Esther SM Wong, John SY Lim, Rafidah A Mutalif, Raju Navasankari, Henry Yang, Yi Y Liow, Thomas Boudier, Graham D Wright, Alan Colman, Brian Burke, Colin L Stewart, Oliver Dreesen |

The funders had no role in study design, data collection and interpretation, or the decision to submit the work for publication.

### Author contributions

AC, OD, Conception and design, Acquisition of data, Analysis and interpretation of data, Drafting or revising the article; PFO, RAM, RN, BD, YYL, Acquisition of data; ESMW, Acquisition of data, Analysis and interpretation of data, Drafting or revising the article, Contributed unpublished essential data or reagents; JSYL, GDW, Acquisition of data, Analysis and interpretation of data; HY, SKS, Analysis and interpretation of data; TB, Conception and design, Analysis and interpretation of data; AC, Conception and design, Drafting or revising the article; BB, CLS, Conception and design, Analysis and interpretation of data, Drafting or revising the article

### Author ORCIDs

Graham D Wright, http://orcid.org/0000-0003-2362-1312

### Ethics

Animal experimentation: This study was performed in strict accordance with the recommendations in the Guide for the Care and Use of Laboratory Animals of the National Institutes of Health. All of the animals were handled according to approved institutional animal care and use committee (IACUC) protocols (140960) of the Institute of Medical Biology, A*STAR, Singapore.

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
