## [Decision Letter]

Thank you for submitting your work entitled “Progerin reduces LAP2α-telomere association in Hutchinson-Gilford Progeria” for peer review at *eLife*. Your submission has been evaluated by James Manley (Senior Editor), Karsten Weis (Reviewing Editor), and two reviewers, one of whom, Claus Azzalin, has agreed to share his identity.

The reviewers have discussed the reviews with one another and the Reviewing Editor has drafted this decision to help you prepare a revised submission.

This study describes a new cell culture model for investigating the molecular mechanisms underlying the progeroid phenotype, by expressing wild-type lamin A or progerin (the mutant form of lamin A expressed in HGPS) under an inducible promoter in normal primary human fibroblasts. This system recapitulates several phenotypes observed in patient cells and other cell culture models, and the authors demonstrate that exogenous TERT rescues most of these phenotypes. Furthermore they show that a well-known binding partner of lamin A, LAP2α, has reduced binding to progerin and begin to investigate the mechanism by which reduced lap2α binding may contribute to progeroid phenotypes.

The reviewers agreed that this manuscript contains interesting information and that the newly developed cell culture model is very useful. However, there was also a consensus that key experiments and experimental details are missing from this study and that significant revisions are needed before the paper could be accepted in *eLife*.

Essential revisions:

1) A key issue is that the authors use different cell types to come to their conclusions. For example, the conclusion that physiological levels of TERT are sufficient to prevent progerin-associated phenotypes is not supported. There are other unique features of embryonic stem cells that could explain this or contribute. Hence the authors need to test their hypothesis more directly, for example by inhibiting telomerase in ES cells.

2) Similarly, it is unclear why the authors switched to patient derived cells to investigate the association of lap2α with telomeres, especially after they highlight the advantages of the cell culture model they developed. Some validation experiments should also be conducted in the progerin inducible system.

3) The phenotypic analysis of lap2α expression on progerin-associated phenotypes is expected to mirror that of TERT in Figure 1 (main and supplement) to really determine whether these proteins operate by the same or different mechanisms. Again it is unclear why the authors are going back and forth between systems but proliferation data are shown in the inducible expression system, whereas heterochromatin data are shown for patient cells. The authors should include experiments addressing how exogenous expression of lap2α affects nuclear morphology, heterochromatin, proliferation, DNA damage foci, senescence, and importantly, telomere length/stability all in the same system. The telomere analysis is particularly obvious, given the data presented in Figure 4. Finally, the controls used in this figure are different. Throughout the paper and in Figure 5, wild-type lamin A is used as a control for progerin. However, the authors switch to a ‘control vector’ in panels C-J. Is this because expression of lamin A alone has some effects?

4) The authors should test whether Lap2α is required for TERT to rescue the progerin-associated phenotypes (and perhaps vice versa, in cells that express endogenous TERT).

Additional points:

1) Figure 1—figure supplement 1, panel H: why is natural Progerin running much closer to LMNC than to LMNA, while it is the opposite for the V5-tagged Progerin (see same figure panel A or Figure 1)? It is unlikely that one V5 tag (10 aa) could explain this discrepancy.

2) The effects of over-expressing LAP2α on H3K27me3 levels need to be tested also in normal fibroblasts. Without this control it is not clear whether LAP2α and progerin control H3K27me3 levels within the same pathway.

3) It would be interesting to add data about the ability of LAP2α to prevent 53BP1 foci accumulation in progerin expressing cells.

4) The experiments described in Figure 5—figure supplement 2 are difficult to interpret and are not helpful. Inhibition of HDACs leads to a plethora of cellular alterations, which are difficult to control. Whether the synergistic/additive effect observed on the growth rates of cells expressing progerin and treated with VPA derive from an ‘open chromatin’ state remains totally unclear. VPA treatment could, for example, lead to de-regulation of one or more transcripts encoding polypeptides responsible for the observed effects. Also, the authors have not directly tested the state of telomeric and subtelomeric heterochromatin but only looked at total H3K27me3 levels. The authors need to tone down their model, which is presented in a rather dogmatic way.

5) The authors neglect to consider in their manuscript the functional interaction between lamin A and the telomere binding protein TRF2 ([61], doi: 10.1038/ncomms6467).

6) Proliferation is quantified in several different ways throughout the manuscript (growth rate (% of control), cell counts over time, growth index, and growth rate (AU), and the methods for these measurements are insufficiently described. It is important that these data are presented consistently across the figures for ease of comparison- the growth index with inset of growth rate seems to be the most useful.

7) Statistical analyses are unclear and inconsistent (for example, some growth rate graphs have asterisks, some do not); each figure legend should describe the statistical analysis of the experiments therein (the actual tests used, not just p-values and n). Throughout the text of the manuscript, it is not clear what trends/differences the authors are claiming to be statistically significant (for example, DNA damage foci in Figure 1 and emPAI in Figure 3—figure supplement 1).

8) The Methods are sparse in detail and do not provide enough information (or references) for someone to reproduce the experiments.

9) Does TERT suppress progerin effects on nuclear morphology (as shown for mESCs in Figure 2)? Also, panels A/B seem more appropriate for supplemental, while a zoom image of nuclear morphology changes would be helpful in the main figure.

10) The authors note in the manuscript text (in the subsection “Super-resolution microscopy reveals impaired localization of LAP2α to telomeres in HGPS cells”) associated with this figure that previous studies have examined the interactome of lamin A and progerin, but they neglect to mention that at least one of these studies identified differential binding partners and LAP2α was not among them ([35], doi:10.4161/nucl.1.6.13512). Also, regarding the in vitro binding studies of LAP2α and wild-type lamin A vs. progerin: the use of the word ‘recombinant’ for proteins generated by in vitro transcription/translation is mis-leading. ‘Recombinant’ suggests the proteins are expressed and purified from bacteria. In this case, there are other proteins present which may affect the interaction. Furthermore, emerin is included as a control without the corresponding input blots or any description in the Methods.

11) Panel E is very confusing and the associated text (in the subsection “Super-resolution microscopy reveals impaired localization of LAP2α to telomeres in HGPS cells”) and figure legend need a simpler explanation (could be supplemental).

12) Panel B seems more appropriate for supplemental, while Figure 5—figure supplement 1 should be in the main figure. Figure 5—figure supplement 2 – this set of experiments is somewhat contradictory to those shown in Figure 2 because ES cells have an ‘open chromatin’ configuration-perhaps cells that have ‘open chromatin’ need to express TERT to protect them from DNA damage? This should be addressed in the text (otherwise, the figure is dispensable).

---

## [Author Response]

*Essential revisions*:

*1) A key issue is that the authors use different cell types to come to their conclusions. For example, the conclusion that physiological levels of TERT are sufficient to prevent progerin-associated phenotypes is not supported. There are other unique features of embryonic stem cells that could explain this or contribute. Hence the authors need to test their hypothesis more directly, for example by inhibiting telomerase in ES cells*.

We agree that this is a very important point and had actually performed part of these experiments at the time of our initial submission. These additional results are presented at the end of the subsection “Progerin impairs proliferation, induces premature senescence and loss of lamin B1 in primary fibroblasts” of the revised manuscript, in the revised Figure 2 and in Figure 2—figure supplement 1:

a) It has been well documented that telomerase is downregulated during ESC differentiation (2). To test whether physiological downregulation of telomerase would render the cells – derived from progerin expressing ESC – susceptible to progerin expression, we differentiated them. We generated embryoid bodies using the hanging drop method, and measured differentiation by analyzing the extent of the differentiated outgrowth upon plating the EBs (Figure 2). As shown in revised Figure 2, expression of lamin A or progerin did not significantly alter the ability of these cells to aggregate and form embryoid bodies (Figure 2). However, as shown in Figure 2, the differentiated cell layer arising from the embryoid bodies after plating was significantly smaller in embryoid bodies expressing progerin (Figure 2). The impaired growth of the differentiated outgrowth layer was accompanied by an increase in 53BP-1 DNA damage foci staining (Figure 2—figure supplement 1).

b) To further determine whether telomerase was responsible for protecting pluripotent mESC from the detrimental consequences of progerin expression, we derived mESC from telomerase deficient mice and expressed progerin in these cells. As shown in revised Figure 2, expression of progerin in pluripotent TERT-deficient mESC led to a dramatic decline in cell numbers (Figure 2) and triggered widespread differentiation of the mESC (Figure 2). Consistent with these findings, progerin expressing TERT-/- mESC formed significantly smaller embryoid bodies (Figure 2).

c) Lastly, as the reviewers pointed out, there could be significant differences in the composition of the nuclear lamina and chromatin organization in mESC versus human fibroblasts. To investigate whether our BioID results can be recapitulated in mESC, we expressed BirA-progerin and BirA-lamin A in pluripotent mESC. In agreement with what we observed in our human fibroblasts model (Figure 3), progerin exhibits a weaker “interaction” with LAP2 in mESC (Figure 3—figure supplement 2).

In conclusion, these experiments support our conclusion that physiological levels of telomerase are sufficient to prevent progerin-induced phenotypes.

*2) Similarly, it is unclear why the authors switched to patient derived cells to investigate the association of LAP2α with telomeres, especially after they highlight the advantages of the cell culture model they developed. Some validation experiments should also be conducted in the progerin inducible system*.

We agree with the reviewers' concern and conducted additional experiments in the progerin inducible system and vice versa.

a) In the original submission, we reported an inverse correlation between progerin levels and H3K27me3 loss in wild type cells (+/- TERT) (Figure 1). We have now recapitulated these findings in two primary HGPS cells as well as a TERT+HGPS line and added these results as additional panels in Figure 1 (panel F) and Figure 1—figure supplement 1). These results are described in the revised manuscript in the subsection “Progerin impairs proliferation, induces premature senescence and loss of lamin B1 in primary fibroblasts”.

b) In the original submission, we observed that LAP2α was localized to telomeres, with a sharp drop of its signal immediately adjacent to the telomere (Figure 4). We have now reproduced these findings in TERT+ wild type cells ectopically expressing progerin (Figure 4—figure supplement 1). Nevertheless, we did not observe a significant shift in LAP2α signal around the telomeres upon progerin expression in this system. This is in contrast to our previous results in HGPS cells, but may not be unexpected: the original analysis (Figure 4) was performed on TERT+HGPS cells that have been exposed to progerin since their inception, whereas our model cells in Figure 4–figure supplement 4H have only been exposed to progerin for 5 days. They may therefore not have had time to develop an equally strong phenotype in this regard. We mentioned these results in the subsection “Super-resolution microscopy reveals impaired localization of LAP2α to telomeres in HGPS cells” of the revised manuscript.

c) We conducted additional key experiments in our progerin inducible system. These experiments are described in point 3 below.

*3) The phenotypic analysis of LAP2α expression on progerin-associated phenotypes is expected to mirror that of TERT in*
Figure 1
*(main and supplement) to really determine whether these proteins operate by the same or different mechanisms. Again it is unclear why the authors are going back and forth between systems but proliferation data are shown in the inducible expression system, whereas heterochromatin data are shown for patient cells*.

As described above, we agree with the reviewer and conducted additional experiments in both cellular systems.

*The authors should include experiments addressing how exogenous expression of lap2α affects*:

Nuclear morphology

This is an important point that we had to clarify. Hence, we systematically investigated the consequences TERT or LAP2α expression on nuclear morphology. We included these results in Figure 1—figure supplement 1 and mentioned these results in the subsections “Progerin impairs proliferation, induces premature senescence and loss of lamin B1 in primary fibroblasts” and “Increased LAP2α specifically rescues progerin-induced defects”.

a) To address whether TERT rescues nuclear morphology defects, we analyzed two different HGPS-patient derived fibroblasts before and after transduction with hTERT, and cultured these cells for an extended period of time. As shown in Figure 1—figure supplement 1, hTERT expression did not ameliorate nuclear abnormalities. These results are consistent with TERT-positive normal dermal fibroblasts expressing progerin – and exhibiting nuclear morphology defects (Figure 1—figure supplement 1).

b) To address whether ectopic expression of LAP2α ameliorates progerin induced nuclear abnormalities, we introduced LAP2α into two independent TERT+HGPS lines and scored nuclear abnormalities before and after doxycycline inducible LAP2α expression. As shown in Figure 1—figure supplement 1, ectopic expression of LAP2α did not ameliorate nuclear defects in HGPS cells. We made the same observation in wild type cells +/- hTERT +/- ectopic LAP2α (data not shown).

Heterochromatin

In the original submission, we showed that expression of LAP2α in TERT+HGPS cells increase H3K27me3 levels (Figure 5). We recapitulated these findings in wild type cells expressing progerin +/- ectopic LAP2α. As shown revised Figure 5, ectopic LAP2α expression increased H3K27me3 levels in progerin expressing wild type cells. This result is described in the subsection “Increased LAP2α specifically rescues progerin-induced defects” of the revised manuscript.

Proliferation

Investigating whether ectopic expression of LAP2α in primary HGPS cells would increase their proliferation rate and or telomere length is an extremely interesting point. Doing such an experiment has nevertheless one major shortcoming that may render its interpretation difficult: as shown in Figure 1—figure supplement 1 (and consistent with published data from the Lansdorp Group: [19]), HGPS cells have dramatically shortened telomeres and prematurely senesce. Elongating telomeres will thus remain a limiting step in alleviating primary HGPS cells defects or performing genetic experiments, regardless of any potential effects from LAP2α.

Nevertheless, we previously transduced HGPS cells with constructs to express LAP2α or hTERT but only obtained clones with the hTERT construct after selection. Although this is no definitive proof, this result suggests that LAP2α does not activate endogenous telomerase, or by some other mechanism elongate telomeres.

DNA damage foci, senescence

We agree this is a very important point. As suggested by the reviewer, we analyzed the number of DNA damage foci (by 53BP-1 staining), and senescent cells (by β-gal staining) in progerin expressing cells +/- LAP2α. As shown in Figure 5—figure supplement 2, co-expression of LAP2α in progerin expressing cells reduced the number of DNA damage foci (as compared to cells expressing only progerin). In addition, we observed a significant reduction in senescence-associated β-gal positive cells. We added these results in Figure 5—figure supplement 2), described these results in the subsection “Increased LAP2α specifically rescues progerin-induced defects*”* and in the Discussion of the revised manuscript.

*[…] and importantly, telomere length/stability all in the same system. The telomere analysis is particularly obvious, given the data presented in*
Figure 4*. Finally, the controls used in this figure are different*.

As mentioned above, our experiments would suggest that LAP2α does not activate telomerase nor increases telomere length. Although somewhat speculative, we think telomere length per se does not appear to matter. Mouse cells have very long (50 kB) telomeres and are similarly susceptible to progerin as human telomeres (see Figure 2 and Figure 2—figure supplement 1).

*Throughout the paper and in*
Figure 5*, wild-type lamin A is used as a control for progerin. However, the authors switch to a ‘control vector’ in panels C-J. Is this because expression of lamin A alone has some effects*?

We did perform a couple of experiments expressing LAP2α in cells overexpressing lamin A as a control (instead of an empty vector construct). Similar to the situation in WT cells, ectopic expression of LAP2α in lamin A expressing cells reduced their proliferation rate. In the same experiment, expression of LAP2α ameliorated progerin-induced proliferation defects (Figure 7). Lastly, since exogenous lamin A expression did not significantly affect the proliferation rate of cells (Figure 1—figure supplement 1), their transcriptome (Figure 1), the senescence associated loss of lamin B1 (Figure 1—figure supplement 1), nor embryoid body formation and differentiation of WT mESC (Figure 2), we used lamin A and vector controls interchangeably depending on the experiments (and vector system) designs.

*4) The authors should test whether LAP2α is required for TERT to rescue the progerin-associated phenotypes (and perhaps vice versa, in cells that express endogenous TERT)*.

We demonstrate that LAP2 is not necessary for TERT to rescue progerin-induced proliferation defects. This result in shown in Figure 5—figure supplement 1. TERT expression prevented progerin-induced proliferation defects in the absence of LAP2 (Figure 5—figure supplement 1). TERT prevented the progerin induced proliferation phenotype regardless of the presence or absence of LAP2.

Additional points:

*1)*
Figure 1—figure supplement 1*, panel H: why is natural progerin running much closer to LMNC than to LMNA, while it is the opposite for the V5-tagged progerin (see same figure panel A or*
Figure 1*)? It is unlikely that one V5 tag (10 aa) could explain this discrepancy*.

This is an interesting point: We have previously noted that even in patient derived HGPS cells, the migration pattern of progerin can differ between different studies and possibly different cell types (Zhang et al., Cell Stem Cell 2011, Figure 3; Liu et al., Nature 2011, Figure 3; McClintock et al., PLoS One 2007, Figure 3). We currently do not know whether progerin may be subject to some additional modifications.

*2) The effects of over-expressing LAP2α on H3K27me3 levels need to be tested also in normal fibroblasts. Without this control it is not clear whether LAP2α and Progerin control H3K27me3 levels within the same pathway*.

We have performed this experiment – it is added in Figure 5—figure supplement 1. Expression of LAP2α in wild type cells did not significantly alter H3K27me3 levels. Thus, LAP2α expression increased H3K27me3 levels in progerin expressing cells with reduced H3K27me3 – but not in wild type cells with normal H3K27me3 levels.

*3) It would be interesting to add data about the ability of LAP2α to prevent 53BP1 foci accumulation in progerin expressing cells*.

Yes, we have added these results as described above (Figure 5—figure supplement 2).

*4) The experiments described in*
Figure 5—figure supplement 2
*are difficult to interpret and are not helpful. Inhibition of HDACs leads to a plethora of cellular alterations, which are difficult to control. Whether the synergistic/additive effect observed on the growth rates of cells expressing progerin and treated with VPA derive from an ‘open chromatin’ state remains totally unclear. VPA treatment could, for example, lead to de-regulation of one or more transcripts encoding polypeptides responsible for the observed effects. Also, the authors have not directly tested the state of telomeric and subtelomeric heterochromatin but only looked at total H3K27me3 levels. The authors need to tone down their model, which is presented in a rather dogmatic way*.

We agree with the reviewer that the interpretation of this experiment is worded too strongly. Given the reviewers comments on this experiment, we would be agreeable to remove the VPA experiment (Figure 5—figure supplement 3) from the manuscript. At this stage, we changed the wording of the experiment / model in the subsection “Increased LAP2α specifically rescues progerin-induced defects*”*: “Based on these results we speculate that a reduction in heterochromatin may render cells more susceptible to progerin-induced proliferation defects.” We also added the following statement to the last paragraph of the Discussion. “However, it remains to be investigated whether progerin affects H3K27me3 specifically at telomeric- or subtelomeric regions, and how expression of LAP2α ameliorates this loss“.

*5) The authors neglect to consider in their manuscript the functional interaction between lamin A and the telomere binding protein TRF2 (*[61]*,* doi: 10.1038/ncomms6467*).*

We agree with the reviewer that the study by Wood et al., is interesting and thought-provoking. The authors of this paper show that telomeres / or ITLs are disrupted in HGPS cells. What remains unclear is whether the “disruption of ITLs in HGPS cells” is a cause of progerin-induced senescence – or a consequence. It would be interesting to see whether ITL disruption occurs in TERT+ HGPS cells – or whether ectopic LAP2α expression would prevent it. In addition, it would be interesting to see whether ITL disruption can be recapitulated in progerin expressing wild type cells. However, we think these experiments are beyond the scope of our study. Lastly, immunoprecipitation experiments with lamins are notoriously difficult to perform, as lamins require high salt concentrations to be solubilized. Having said this, we were unable to detect TRF2 in the BioID interactome of lamin A.

6) Proliferation is quantified in several different ways throughout the manuscript (growth rate (% of control), cell counts over time, growth index, and growth rate (AU)), and the methods for these measurements are insufficiently described. It is important that these data are presented consistently across the figures for ease of comparison- the growth index with inset of growth rate seems to be the most useful.

We agree and have now added the “growth index” as an inset in each figure panel: Figure 1—figure supplement 1, Figure 5, Figure 5, Figure 5, Figure 5—figure supplement 1.

*7) Statistical analyses are unclear and inconsistent (for example, some growth rate graphs have asterisks, some do not); each figure legend should describe the statistical analysis of the experiments therein (the actual tests used, not just p-values and n). Throughout the text of the manuscript, it is not clear what trends/differences the authors are claiming to be statistically significant (for example, DNA damage foci in*
Figure 1
*and emPAI in*
Figure 3—figure supplement 1*).*

We agree and added details regarding the type of statistical analysis we used in every figure legend.

8) The Methods are sparse in detail and do not provide enough information (or references) for someone to reproduce the experiments.

We added a subsection to the Materials and methods on the culture of mESC and the derivation of TERT-/- mESC (“Mouse embryonic stem cell derivation, culture and differentiation*”*).

*9) Does TERT suppress progerin effects on nuclear morphology (as shown for mESCs in*
Figure 2*)?*

We addressed this point in point 3 of the major comments and added the results to Figure 7. Our data do not provide any evidence that TERT suppresses progerin induced nuclear morphology defects, regardless as to whether we expressed TERT in HGPS cells, progerin in TERT-positive fibroblasts or undifferentiated mESC. In addition, the nuclear architecture of mESC appears to be a lot more plastic than in human fibroblasts, and present a high degree of what would be qualified as abnormal nuclei in somatic cells.

Author response image 1.**DOI:**
http://dx.doi.org/10.7554/eLife.07759.020

Also, panels A/B seem more appropriate for supplemental, while a zoom image of nuclear morphology changes would be helpful in the main figure.

We agree and moved these panels into Figure 2—figure supplement 1.

*10) The authors note in the manuscript text (in the subsection “Super-resolution microscopy reveals impaired localization of LAP2α to telomeres in HGPS cells”) associated with this figure that previous studies have examined the interactome of lamin A and progerin, but they neglect to mention that at least one of these studies identified differential binding partners and lap2α was not among them (*[35]*,*
*doi:10.4161/nucl.1.6.13512**).*

Kubben and colleagues used a high affinity OneSTrEP-tag and immunoprecipitation while our system identifies neighbors based on physical proximity in living cells. Both techniques have specific advantages, and it is possible that each of them is more appropriate at characterizing different types of interactions. Thus, we feel it is difficult to directly compare the dataset of Kubben et al. with ours. In addition a number of papers from the Foisner group identified LAP2α as an interactor of lamin A ([16]; Markiewicz et al., 2002; [45]). However, it remained unclear whether progerin exhibited an impaired interaction with LAP2α.

Also, regarding the in vitro binding studies of LAP2α and wild-type lamin A vs. progerin: the use of the word ‘recombinant’ for proteins generated by in vitro transcription/translation is mis-leading. ‘Recombinant’ suggests the proteins are expressed and purified from bacteria. In this case, there are other proteins present which may affect the interaction. Furthermore, emerin is included as a control without the corresponding input blots or any description in the Methods.

Indeed, we thank the reviewers for pointing this out. We changed the text in the subsection “BioID analysis reveals an impaired interaction between LAP2α and progerin*”* of the revised manuscript. We also added a different western blot picture to the revised Figure 3, which includes the emerin input.

11) Panel E is very confusing and the associated text (in the subsection “Super-resolution microscopy reveals impaired localization of LAP2α to telomeres in HGPS cells”) and figure legend need a simpler explanation (could be supplemental).

We agree and added more detail to the subsection “Super-resolution microscopy reveals impaired localization of LAP2α to telomeres in HGPS cells“ of the revised manuscript. As the reviewer requested, we moved these results to the Figure 4—figure supplement 4F.

*12) Panel B seems more appropriate for supplemental, while*
Figure 5—figure supplement 1
*should be in the main figure.*

We rearranged Figure 5 and added additional panels to include histone data on wild type cells as requested above. To prevent overcrowding Figure 5, we left what used to be Figure 5—figure supplement 1, in Figure 5—figure supplement 1. We hope this is fine.

Figure 5—figure supplement 2
*– this set of experiments is somewhat contradictory to those shown in*
Figure 2
*because ES cells have an ‘open chromatin’ configuration-perhaps cells that have ‘open chromatin’ need to express TERT to protect them from DNA damage? This should be addressed in the text (otherwise, the figure is dispensable).*

As mentioned above, we would be agreeable with removing the VPA experiment (now Figure 5—figure supplement 3) from the manuscript.